# Modelling the species-area relationship using extreme value theory

**Luís Borda-de-Água** [1,2,3,8] ✉, **M. Manuela Neves**[4], **Luise Quoss** [5,6],
**Stephen P. Hubbell**[7], **Filipe S. Dias** [1,2,3,8] & **Henrique M. Pereira** [1,2,3,5,6]

The nested species-area relationship, obtained by counting species in increasingly larger areas in a nested fashion, exhibits robust and recurring qualitative and quantitative patterns. When plotted in double logarithmic scales it shows three phases: rapid species increase at small areas, slower growth at intermediate scales, and faster rise at large scales. Despite its significance, the theoretical foundations of this pattern remain incompletely understood. Here, we develop a theory for the species-area relationship using extreme value theory, and show that the species-area relationship is a mixture of the distributions of minimum distances to a starting sampling focal point for each individual species. A key insight of our study is that each phase is determined by the geographical distributions of the species, i.e., their ranges, relative to the focal point, enabling us to develop a formula for estimating the number of species at phase transitions. We test our approach by comparing empirical species-area relationships for different continents and taxa with our predictions using Global Biodiversity Information Facility data. Although a SAR reflects the underlying biological attributes of the constituent species, our interpretations and use of the extreme value theory are general and can be widely applicable to systems with similar spatial features.

Larger areas tend to harbour a larger number of species. The species-area relationship (SAR) describes how the number of species increases as a function of the size of the area. The nested SAR, obtained by counting species in increasingly larger areas, exhibits consistent qualitative and quantitative patterns across different taxa and habitats, a generality that makes it one of the most fundamental patterns in ecology[1] and a subject of extensive study in theoretical ecology[2–6]. However, the term 'SAR' encompasses various methods of data collection and analysis[7–10]. Here, we exclusively focus on cases where species are enumerated within nested areas (Fig. 1).

When a SAR is plotted across a wide range of areas on a double logarithmic plot, it displays three distinct regions, forming a triphasic curve[6,11–14]. Initially, for small areas, which we call Phase I, the number of species increases rapidly, with a slope approaching 1. Subsequently, for intermediate area sizes, Phase II, the SAR exhibits slower growth, following a power law with an exponent smaller than one. Finally, for large area sizes, Phase III, the SAR experiences rapid growth again, with a slope tending to one[15].

Several explanations have been proposed for each phase of the SAR. Williamson[16], building on an argument by Plotkin and Levin[17],

[1]CIBIO/InBio, Centro de Investigação em Biodiversidade e Recursos Genéticos, Laboratório Associado, Universidade do Porto; Campus Agrário de Vairão, 4485-661 Vairão, Portugal. [2]CIBIO/InBio, Centro de Investigação em Biodiversidade e Recursos Genéticos, Laboratório Associado, Instituto Superior de Agronomia, Universidade de Lisboa; Tapada da Ajuda, 1349-017 Lisbon, Portugal. [3]BIOPOLIS Program in Genomics, Biodiversity and Land Planning, CIBIO; Campus de Vairão, 4485-661 Vairão, Portugal. [4]Instituto Superior de Agronomia and Centro de Estatística e Aplicações, Universidade de Lisboa (CEAUL); Tapada da Ajuda, 1349-017 Lisbon, Portugal. [5]German Centre for Integrative Biodiversity Research (iDiv) Halle-Jena-Leipzig; Puschstraße 4, 04103 Leipzig, Germany. [6]Institute of Biology, Martin Luther University Halle-Wittenberg; Am Kirchtor 1, 06108 Halle (Saale), Germany. [7]Department of Ecology and Evolutionary Biology, University of California Los Angeles, Los Angeles, CA, USA. [8]Present address: Centro de Ecologia Aplicada Professor Baeta Neves (CEABN), Instituto Superior de Agronomia, 1349-017 Tapada da Ajuda, Lisbon, Portugal. ✉e-mail: lbagua@gmail.com

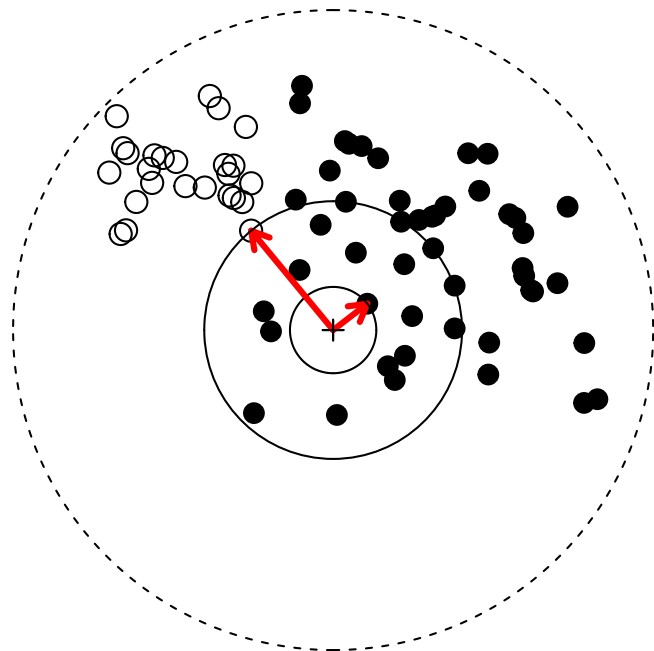

**Fig. 1 | Sampling scheme illustration for two hypothetical species.** Open and closed dots symbolise the hypothetical distributions of individuals of two species. The red arrows indicate the location of the nearest individual to the focal point, marked by a cross in the centre of the figure. The location of the closest individual provides the information to construct the SAR, with the circles identifying the corresponding areas. The spatial distributions exemplify two scenarios: one where a species' range encompasses the focal point (closed dots) and another where it does not (open dots).

attributed the steep slope of Phase I to species occurring at very low densities in very small areas. In the limiting case where species are represented by only one individual, this leads to a SAR with a slope of one on a double logarithmic plot. For Phase II, Preston[3] explained the power law relationship, with a presumed exponent approximately equal to 0.25, by assuming an underlying lognormal species abundance distribution under the canonical assumption that the maximum of the distribution of individuals coincides with the most abundant species. Subsequently, May[5] demonstrated that the slope is still approximately equal to 0.25 when the canonical approach is relaxed. Finally, the steep slope characterising Phase III has been explained as the result of the combination of species from different biogeographical realms[6,18] or, similarly, due to the sampled areas being of similar magnitude to the geographic ranges of species[19,20].

However, a unified theory explaining the three phases and their transition points has been missing[21]. This is particularly important, as the nested SAR describes the first wave of extinctions after habitat loss corresponding to the loss of endemics[22,23]. Therefore, projections of biodiversity loss in response to land-use change are contingent on having a theoretical understanding of this pattern. Here, we use order statistics and extreme value theory, also known as statistics of extremes, to construct a mathematical framework of the SAR, aiming to provide a unified explanation of its key characteristics. Order statistics deals with the properties of ordered random variables and, in particular, with the minimum and maximum values[24], and extreme value theory examines their asymptotic behaviour for large sample sizes[25–27]. These methods are pertinent in our context because SARs are obtained from encountering the first individual of each species in the community, assuming that sampling initiates from a specified focal point, as depicted in Fig. 1[28]. Recognising that the first individual of a species encountered is the one with the *minimum* distance to the focal point, obtaining a SAR corresponds to identifying the minimum value of the distribution of distances for each species in the

community (and converting this distance to an area). Here, we develop a theoretical framework for the SAR using extreme value theory and demonstrate that it provides a unified explanation for the SAR phase transitions. We validate our predictions with empirical data from diverse taxa and continents, showing that our findings have broad applicability and suggest that similar statistical principles may underlie other spatial patterns.

## Results

### Extreme value theory

Let $F_i(r)$ be the cumulative distribution function (cdf) of the distances, $R_i$, of individuals of a species $i$ to a focal point. Then, the cdf of the corresponding minimum, $L_i(r)$, for a sample of $n$ individuals is simply the complement of the probability of all individuals occurring at a distance larger than $r$, that is,

$$L_i(r) = 1 - \left(1 - F_i(r)\right)^n \tag{1}$$

(e.g., ref. 24). $L_i(r)$ can also be expressed in terms of the area, $L_i(A)$, considering that $A = \pi r^2$. Recognising that $L_i(A)$ corresponds to the probability of a species contributing to the SAR for a given area $A$, then a mixture, $S_M(A)$, can be defined as an equally weighted sum, with weights equal to $1/S_T$, of the distributions of the minima of all species, $S_T$, in the community

$$S_M(A) = \frac{1}{S_T} \sum_{i=1}^{S_T} L_i(A) \tag{2}$$

Thus, under this formulation, $S_M(A)$ is a cdf, corresponding to the probability of having found a new species, and it can be interpreted as the proportion of species of the community observed in a given area $A$. The derivative of $S_M(A)$, $S'_M(A)$ is, then, the probability density of finding a new species for an area $A$.

When the number of individuals, $n$, is large, extreme value theory shows that the number of the possible distributions of minima, $L(A)$, can, after a suitable normalisation, be narrowed down to three, the Weibull, the reverse Fréchet and the reverse Gumbel distributions[25–27]. Moreover, these distributions can be combined into a single distribution, known as the generalised extreme value (GEV) distribution for minima, with location $\mu$, scale $\sigma$ and shape $\zeta$ (refer to Supplementary Note 1 for details). Deriving the parameters of the distribution of minima, $L(A)$, i.e., the GEV, from those of the parent distribution, $F(r)$, proves to be a daunting, if not impossible, task in most cases (see Supplementary Note 1). Here we use simulations to elucidate the relationship between the relative position of the range for a species relative to the initial point of sampling (the focal point) and how it contributes to the SAR. Although these simulations are based on simplistic assumptions, they will guide in establishing heuristic guidelines for predicting the transitions among phases in real-world SARs.

### Computer simulations

Simulations were conducted using 15,000 species, each with 1000 individuals (Fig. 2). For each species, the locations of the individuals were assumed to conform to isotropic bivariate normal distributions with the same variance, $\sigma_N^2 = 1$, but with centres $(x_0, y_0)$ presumed to be uniformly randomly distributed. Consequently, the distribution of distances to a focal point at (0,0) follows a Rice distribution with location parameter $\upsilon_p = \sqrt{x_0^2 + y_0^2}$, and scale parameter $\sigma_p = \sigma_N$ (e.g., ref. 29). For instance, the black dots in Fig. 2a represent the SAR obtained from one realisation of this sampling scheme by identifying the minima of the distribution of distances of each species. To estimate the parameters of the corresponding GEV, we repeat this procedure 1000 times, yielding a set of 1000 minima for each species. Once the

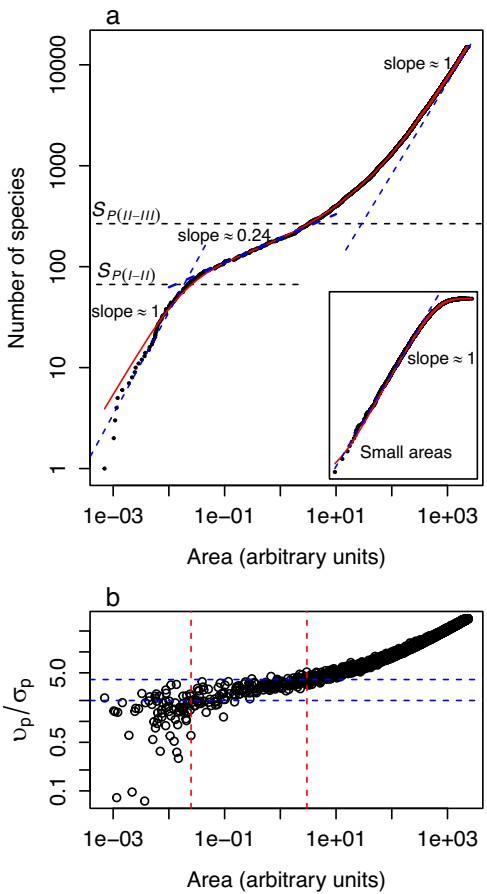

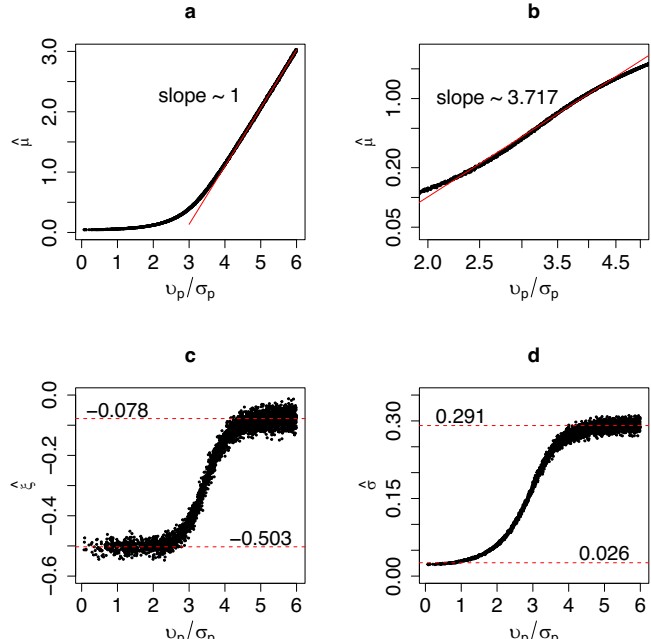

**Fig. 3 | The estimated parameters of the generalised extreme value distribution (GEV), as a function of $v_p$, the location parameter of the (parent) Rice distribution. a** The location parameter of the GEV, $\hat{\mu}$. **b** A zoom in of $\hat{\mu}$ in double logarithm axes for the values of $v_p$ associated with Phase II. **c** The shape parameter, $\hat{\xi}$. **d** the scale parameter, $\hat{\sigma}$. The red dashed lines in plots (**c**, **d**) correspond to the mean of the values for $v_p > 5$, upper line, and for $v_p < 1$, lower line. Recall that $v_p$ is also the distance of the centre of the bivariate normal distribution of the individuals' locations to the focal point.

**Fig. 2 | A triphasic nested species-area relationship (SAR). a** SAR obtained with simulations assuming 15,000 species with individuals distributed according to bivariate normal distributions with scale parameter $\sigma_p = 1$ and randomly uniform centres. The black dots correspond to one realisation where we sampled 1000 individuals from each species and identified the minimum. This procedure was repeated $N = 1000$ times, from which the parameters of the generalised extreme value distribution were estimated and the red line was then obtained using expression (2). The dashed blue lines were obtained by fitting the SAR using linear regressions for small, intermediate, and large areas. The inset shows the SAR obtained from simulations for small area sizes only, assuming $\sigma_p = 1$, and the centres uniformly randomly distributed within a radius region equal to 2 (arbitrary units). **b** The ratio $v_p/\sigma_p$ of the parameters of the (parent) Rice distribution for each black point in plot a, as a function of the area. The horizontal dashed lines correspond to the ratios $v_p/\sigma_p = 2$ and 4. Species contributing to Phase I primarily have ratios $\frac{v_p}{\sigma_p} < 2$, those contributing to Phase II have $2 < \frac{v_p}{\sigma_p} < 4$ and those contributing to Phase 3 have $\frac{v_p}{\sigma_p} > 4$.

GEVs for all species were obtained, we generated the red line shown in Fig. 2a by applying expression (2). Whether by plotting the minima of a single realisation or using expression (2), all three phases are distinctly reproduced (Fig. 2a). The apparent deviation between the realisation and the theoretical prediction for very small areas is due to the low number of individuals. When a similar simulation is done just focusing on small areas, the deviation disappears (Fig. 2a inset). In addition to the previous simulations, we also performed further simulations across a wider range of scenarios, including (i) species with varying range sizes, (ii) spatially non-uniformly distributed ranges and (iii) species with different number of individuals and range sizes; the results are presented in Supplementary Note 4 and Fig. S3–S10. While the main attributes of the SARs and their interpretation through the EVT do not change, these simulations provide further insights into the patterns observed in empirically observed SARs (see Supplementary Note 6). Finally, we conducted simulations assuming the individuals'

locations follow isotropic bivariate Cauchy distributions. The results were qualitatively similar: we observed three phases, with Phases I and III characterised by slopes close to 1 in a log-log plot, and Phase II well approximated by a power law with a slope smaller than 1, but extending over a wider range of areas compared to the simulations using isotropic bivariate normal distributions (see Supplementary Note 4 and Fig. S11).

The key insight gleaned from the simulations is that species situated at varying distances from the focal point contribute to different phases of the SAR. To elucidate this point, for each minimum represented by the black dots in Fig. 2a, we recorded $v_p$ and $\sigma_p$ of the corresponding distribution of distances, and plotted the ratio $v_p/\sigma_p$ as a function of the area (Fig. 2b). It is apparent that the species contributing to Phase I are those that typically exhibit a ratio $v_p/\sigma_p < 2$. Recalling the definitions of $v_p$ and $\sigma_p$, a ratio of $v_p/\sigma_p < 2$, or $v_p < 2\sigma_p$, indicates the bulk of the range of the distribution includes the focal point or, in other words, the focal point is not far from the centres of the distribution. Conversely, the species contributing to Phase III are characterised by $v_p/\sigma_p > 4$, or $v_p > 4\sigma_p$ suggesting the ranges for these species are unlikely to include the focal point, that is, they are far from it. Phase II emerges as an intermediate situation where $2 < v_p/\sigma_p < 4$.

The same conclusions can also be drawn by examining the relationship between the ratio $v_p/\sigma_p$ and the corresponding parameters of the GEV distribution (Fig. 3). Beginning with the location parameter of the GEV, $\mu$, it is apparent from Fig. 3a that the estimated value, $\hat{\mu}$, approaches zero when $v_p/\sigma_p < 2$, suggesting that the minimum, i.e. the first individual detected, is very close to the focal point. On the other hand, for $v_p/\sigma_p > 4$ the relationship between $\hat{\mu}$ and $v_p/\sigma_p$ becomes linear with a slope of one, i.e. the location of the minimum $\hat{\mu}$ becomes directly proportional to the distance between the focal point and the centre of the species range ($v_p$). In fact, when $v_p/\sigma_p \gg 1$, we can assume that the stochastic nature of the minima can be ignored because the

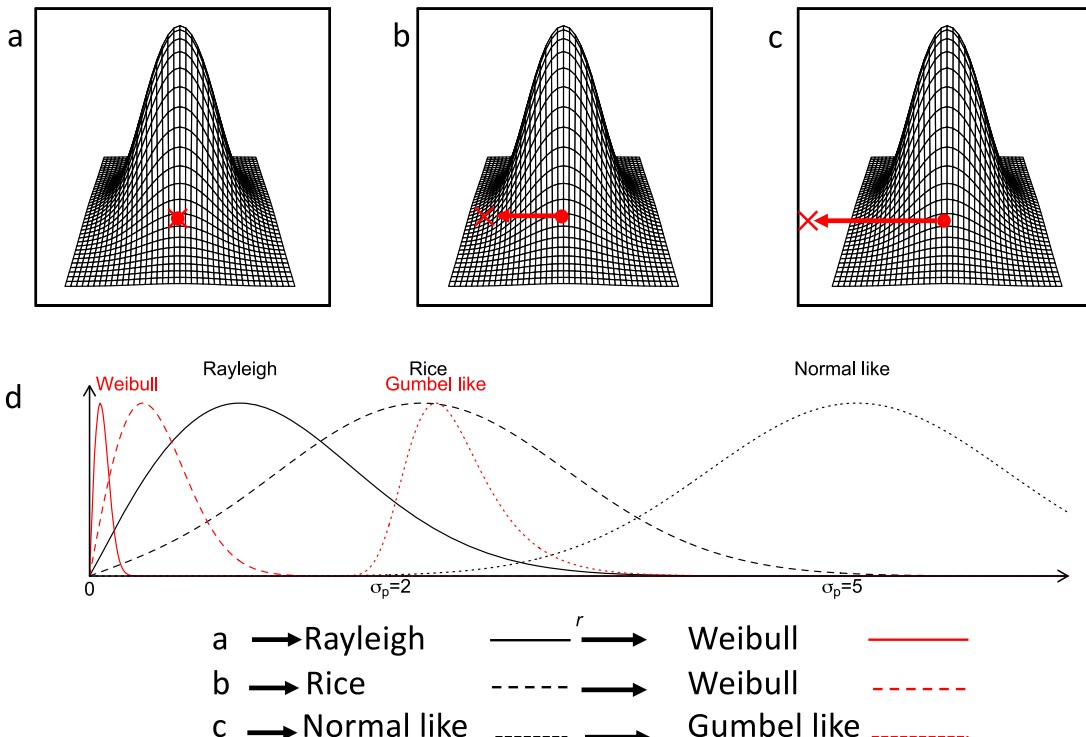

**Fig. 4 | A visual summary illustrating the relationship between the location of a species range relative to the focal point and its contribution to a phase of the SAR.** The top row comprises three plots exemplifying bivariate normal distributions at varying distances from the focal point, represented by the red 'x'. **a** The distribution is centred on the focal point; **b** The focal point falls within a distance of the centre of distribution where $2<\sigma_p^2<4$, where $\sigma_p^2$ is the variance; **c** The focal point is situated at a distance larger than $\sigma_p>4$ (refer also to Fig. 1). **d** The black curves illustrate the corresponding distributions of distances: a Rayleigh distribution for the distribution of plot (**a**), and Rice distributions for the distributions of plots (**b**, **c**). The normal distribution serves as a good approximation for the Rice distribution when the location parameter is significantly larger than its scale parameter (distribution on (**c**)). The red curves represent the respective limiting distributions of the minima: Weibull distributions for the Rayleigh and Rice distributions, and a Gumbel-like distribution when the Rice becomes normal-like. Note that the distributions are not drawn to scale.

plausible range of values of the distribution of the minimum distance is very small compared to the distances to the focal point. In this case, we can describe the ranges of each species simply as discs with a constant radius. Furthermore, in this case and in the limit of large areas, it can be shown that $S = cA$, where $c$ is a constant (refer to Supplementary Note 2; Figs. S1 and S2 for details).

Within the range of $2<v_p/\sigma_p<4$, and in a double logarithmic plot, the relationship between $\hat{\mu}$ and $v_p/\sigma_p$ exhibits an almost power-law behaviour, $\mu \propto \left(\frac{v_p}{\sigma_p}\right)^\beta$, with $\beta \approx 3.45$ (Fig. 3b). This observation, combined with the fact that $v_p$ is uniformly randomly distributed, leads to a power law SAR with exponent $z \approx 0.29$ (see also Supplementary Note 3). This value is similar to that previously mentioned in ref. 3, 5 for Phase II. However, contrary to the assumption of an underlying lognormal distribution for the species abundance distribution used by these authors, in our simulations, all species have the same number of individuals.

The interpretation regarding the location of a species' range relative to the focal point and the phase of the SAR to which it contributes is strengthened by observing the changes in the estimated shape parameter, $\hat{\xi}$, and scale parameter, $\hat{\sigma}$ (Figs. 3c, d). The estimated shape parameter, $\hat{\xi}$, is approximately equal to $-0.5$ when $v_p/\sigma_p<2$, undergoes a transition at $2<v_p/\sigma_p<4$, and then remains close to zero for $v_p/\sigma_p>4$. A value of $\hat{\xi} \approx -0.5$ is compatible with a distribution of minima following a Weibull distribution, which aligns with the expectation that the centre of the range is at or near the focal point, that is, when the distribution of distances of individuals to the focal point follows a Rayleigh distribution (Fig. 4). Furthermore, it can be demonstrated that in this scenario, the number of species, $S$, increases

approximately linearly with the area, $S = cA$; thus, the SAR has a slope of 1 in double logarithmic plots (see Supplementary Note 1). On the other hand, when $v_p/\sigma_p > 4$, then $\hat{\xi}$ is still negative but approaches zero ($\hat{\xi} \lesssim 0$) which is characteristic of a distribution of minima following a Gumbel distribution. This behaviour is expected if the distribution of distances follows, or approximates, a normal distribution (see Fig. 4).

## The number of species at the transitions

The preceding observations offer insight into predicting the proportion of species at which the transitions between phases occur. Beginning with the transition between Phases I and II, the condition $v_p/\sigma_p \lesssim 2$ implies that the species whose centres lie within a distance smaller than $2\sigma_p$ of the focal point contribute to Phase I. If the centres of the ranges for each species are uniformly randomly distributed, and if we denote then the number of the species with centres inside the circle of radius $2\sigma_p$ of the focal point by $S_{P(I-II)}$, then the corresponding proportion of species is given by $S_{MP(I-II)} = \frac{S_{P(I-II)}}{S_T} = \frac{\pi(2\sigma_p)^2}{A_T}$. Thus, once $S_{P(I-II)}$ species have been observed, the subsequent species detected are those contributing to Phase II. In other words, $S_{P(I-II)}$ marks the transition from Phase I to Phase II. Employing a similar reasoning but with condition $v_p/\sigma_p \lesssim 4$, leads to the proportion of species $S_{MP(II-III)} = \frac{S_{P(II-III)}}{S_T} = \frac{\pi(4\sigma_p)^2}{A_T}$, signalling the onset of Phase III. $S_{P(I-II)} = S_T S_{MP(I-II)}$ and $S_{P(II-III)} = S_T S_{MP(II-III)}$ are depicted by horizontal lines in Fig. 2a.

The simulations were performed assuming that all species' ranges were of the same size and spatially uniformly randomly distributed. In

more general terms, the problem can be formulated as follows. Let $f_{v_p}(r,\phi)$ be the probability density function of the spatial distribution of the ranges' centres, $v_p$. Then, given a segment of a ring of width $dr$ and angle $d\phi$ at a distance $r$ from the focal point, the proportion of species within it potentially contributing to a phase corresponds to those whose range centres, $v_p$, fall within the segment of the ring $f_{v_p}(r,\phi)d\phi dr$. However, from these, only the species whose ranges have width $w = 2\sigma_p$ or $w = 4\sigma_p$ larger than $r$ will contribute to $s_{P(I-II)}$, or $s_{P(II-III)}$, respectively. This fraction of species is given by the probability $\mathbb{P}(w > r|, r, \phi)$. Therefore, the proportion of species at a transition is given by

$$S_{MP} = \frac{S_P}{S_T} = \int_0^{R_T}\int_0^{2\pi} \mathbb{P}(w > r|, r, \phi) f_{v_p}(r,\phi)d\phi dr, \quad (3)$$

where $R_T$ the radius of the entire region. When $w = 2\sigma_p$, or $w = 4\sigma_p$, we retrieve $s_{P(I-II)}$ or $s_{P(II-III)}$, respectively (in Supplementary Note 5 we elaborate on this formula for specific cases and discuss the properties of $S_{MP}$; see, in particular, Fig. S12).

The interpretation of $S_{P(I-II)}$ and $S_{P(II-III)}$ warrants some considerations. The phases identified in the SAR in a log-log plot, usually by visual inspection, correspond to regions with distinct slopes. Our simulations have revealed that the transitions between phases correspond to changes in the values of the parameters of the distributions of the minima that occur at approximately $v_p/\sigma_p \approx 2$ and $v_p/\sigma_p \approx 4$. These values guided the derivation of the criteria for estimating $S_{P(I-II)}$ and $S_{P(II-III)}$. However, the transitions between phases are not abrupt, SARs tend to exhibit smooth curves. Consequently, there is always an inherent arbitrariness when choosing the cut-off value between phases, and the criteria $v_p/\sigma_p \lesssim 2$ and $v_p/\sigma_p \lesssim 4$ should be regarded as heuristic. Nevertheless, as we will see, these simple criteria yield satisfactory results when analysing empirical SARs.

### Empirical species-area relationships

Identifying the three phases of a SAR in a real-world setting requires species richness data spanning a broad range of spatial scales. Until recently, single data sets covering all three scales were unavailable. Here, we use data from the Global Biodiversity Information Facility (GBIF), the world's largest biodiversity aggregator of biodiversity records, to illustrate triphasic SARs and employ the insights derived from the simulations to predict the number of species at which the transitions among phases occur.

We generated SARs by first randomly placing the focal point and then determining the shortest distance to the continental coast, which determines the maximum area covered by the SAR. We only considered SARs that had at least 50 species (see 'Methods' for details). However, analysing the SARs empirically obtained from GBIF data requires mentioning two caveats. First, sudden spikes in species counts occur frequently; this happens when the records of the location of several species are aggregated to a single location (e.g., because they were collected in an intensively surveyed site), resulting in SARs with artificially steep slopes. Second, not all SARs exhibit three phases, some only manifest the first two; this is particularly prevalent when SARs cover a limited range of areas, such as those with the focal point located near the coastline or, as we will see below, for those of taxa, such as birds, that have large ranges.

We analysed data for five landmasses (Africa, Australia, Eurasia, North America and South America) and four taxa (Amphibians, Birds, Mammals and Reptiles). In the Supplementary Note 6, we present results for all landmasses and taxa, but here we focus on amphibians and birds from Australia and North America for two main reasons. First, both Australia and North America have the most complete biodiversity assessments, and both have large contiguous areas from

which we can increase the sampling area without intersecting the coasts. Second, amphibian and bird species have the smallest and the largest ranges, respectively (see Fig. S14); thus, we anticipated observing qualitatively very distinct SARs.

For each taxon and landmass, we obtained 200 SARs, the grey lines in (Fig. 5 and Supplementary Note 6, Fig. S13). Note from Figs. 5 and S13 the huge variability across the spatial scales of individual SARs, as expected due to SARs being influenced by both the density of the data in the different sampled areas, and the heterogeneous distribution of species densities. In contrast to other taxa, SARs for bird species do not show a clear Phase III. This was expected, given that Phase III arises when sampling encompasses species from different biogeographical regions with disconnected ranges, but since bird species can have large ranges, Phase III never occurs. Additionally, we derived an average SAR from the 200 sampled SARs (see 'Methods'), depicted by the black dots in Figs. 5 and S13. These average SARs demonstrate the expected trends, namely the different phases. To estimate the number of species at which the transitions between phases occur, $S_{P(I-II)}$ and $S_{P(II-III)}$, we calculated, for each species, the distances of the individuals to the centre of their range using the empirical coordinates of the individuals. The range centres were calculated as the 'centres of gravity' of the individuals' locations. From these distributions, we obtained the standard deviation, $\sigma_p$, for each species. Then, using the heuristic values of $2\sigma_p$ or $4\sigma_p$, as suggested from the simulations, we determined the number of species where the sampling focal point was within $2\sigma_p$ or $4\sigma_p$ from the centre of the range, thus estimating $S_{P(I-II)}$ and $S_{P(II-III)}$, respectively. The values of $S_{P(I-II)}$ and $S_{P(II-III)}$ are displayed in (Fig. 5 and Supplementary Note 6), showing good agreement with the regions where the SARs exhibit inflections. The slopes for each phase, using $S_{P(I-II)}$ and $S_{P(II-III)}$ as approximate delimiters, follow the expected trend, with Phase II having a smaller slope compared to the other two phases; Figs. 5 and S13.

### Discussion

Although the SAR is a basic pattern with a long and venerable history in ecological studies[1], it remains an active area of research[21]. Here, we show that extreme value theory offers a comprehensive framework for analysing the SAR. The power of the EVT lies in its ability to provide several analytical tools and conceptual insights for interpreting the characteristics of the SAR at different spatial scales. For instance, under certain idealised conditions, the steep slopes observed for small spatial scales (Phase I) and large spatial scales (Phase III) can be derived and interpreted using EVT. Our work is a first incursion into the relationship between EVT and an ecological pattern, the SAR, and we anticipate that future applications of EVT and order statistics in ecology will uncover new patterns and provide tools for their quantification, potentially linking these patterns to their underlying processes. Probably, one of the most significant insights from our work is the discovered relationship between a species' range and location and the SAR phase to which it contributes. This observation led to the development of a simple rule of thumb for predicting the number of species at which transitions among the phases of the SAR occur ($S_{P(I-II)}$ and $S_{P(II-III)}$); see also Supplementary Note 5. This rule may be further refined as more detailed information on species ranges is collected. We foresee that $S_{P(I-II)}$ and $S_{P(II-III)}$ may be used in conservation studies aimed at estimating species richness and identifying the area sizes where changes in the rate at which new species appear are expected.

It has long been recognised that nested SARs exhibit three phases when data are collected across a broad range of spatial scales[1,5,6,12,13,16]. Among the three phases, Phase I is probably the one that has received less attention [but see in refs. 16,30]. However, our results show the ubiquity of its presence, therefore it should not be ignored when analysing SARs. This phase is characterised by a steep slope, that can be understood by assuming that species contributing to it have ranges

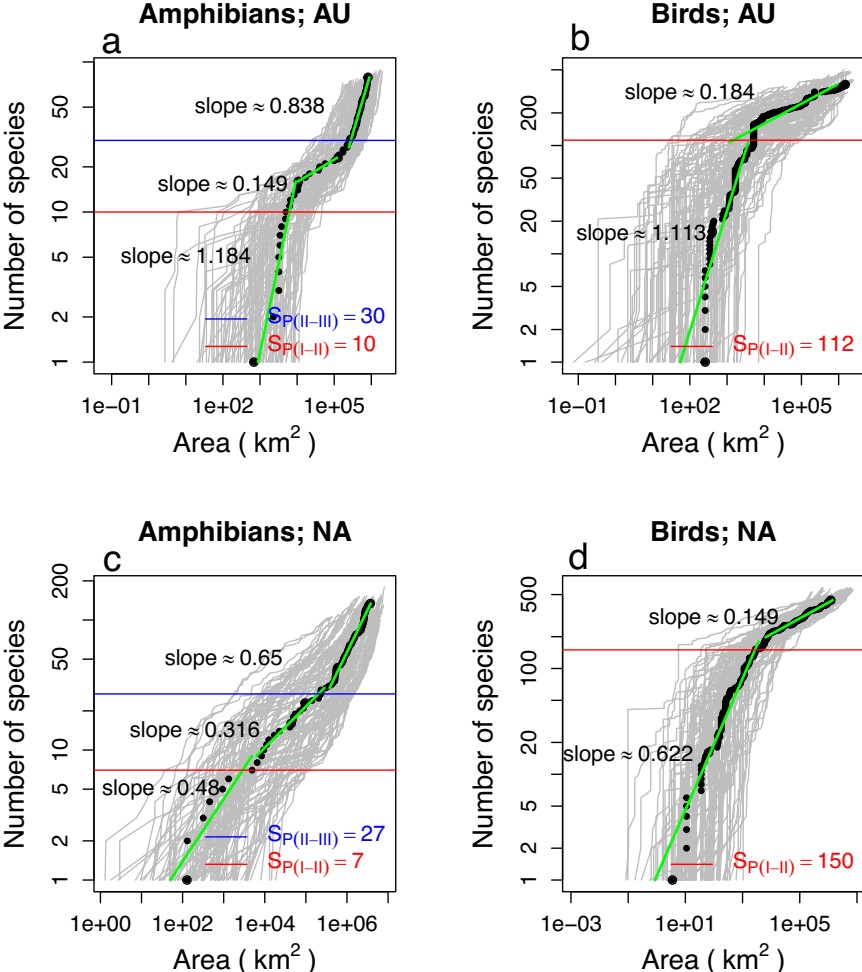

**Fig. 5 | Empirical GBIF SARs alongside corresponding $S_P$ predictions. a, b** SARs for amphibians and for birds from Australia (AU). **c, d** SARs for amphibians and for birds from North America (NA). The grey curves in the background represent SARs obtained from 200 randomly located focal points, while the black dots indicate a mean SAR (see 'Methods' for details). The horizontal red lines represent the number of species predicted at the transition between Phases I and II, $S_{P(I–II)}$, and the horizontal blue lines represent the predicted number between Phases II and III, $S_{P(II–III)}$.

encompassing the sampling focal point (see Supplementary Note 1). On the other hand, Phase II is probably the best documented and analysed[1,3,5,21]. According to our analyses using extreme value theory, this phase corresponds to a transition in the properties of the distributions of minima. In the future, more comprehensive datasets with precise information regarding individual locations may reveal additional properties of this phase, in particular, the determinants of the range of slope values. Finally, Phase III, which requires datasets covering large areas, has been reported less frequently. This phase was originally hinted at by Williams[11] and has since then been reported by other authors[12,13,19] and retrieved in simulations[31]. This phase too can be understood in terms of the location of the species' ranges in relationship to the focal point, specifically, those species whose ranges do not include the focal point (see also Supplementary Note 2).

Rosenzweig[1] associated Phases I, II and III to 'tiny pieces of single biotas', to 'larger pieces of single biotas' and 'areas that have had separate evolutionary histories', respectively. Hubbell[6] called them 'local', 'regional' and 'continental to intercontinental', respectively, suggesting a likely correspondence with these geographical scales. Indeed, the spatial scales at which these phases appear, as well as the observed transitions, likely reflect the spatial scales at which the processes generating diversity (such as speciation, extinction, and migration) operate. However, as we showed, the qualitative features of the SARs, such as the presence of three phases, can be understood

solely in terms of the range sizes of the species and their relative distances to the starting point of the sampling scheme. The nested SAR thus emerges as a specific instance of a broader pattern, one where the observation of the first instance holds decisive significance. That is, the functional form of the nested SAR is not idiosyncratic to ecological communities but rather arises of general 'laws acting around us', borrowing an expression by J. Harte[32]. It is the numerical values of a specific SAR parameter that reflect the underlying biology of the species and the interactions among themselves and with the environment. This suggests that the findings of our work can be applied to any systems where the components form spatial aggregates similar to those observed among species in communities.

## Methods
### The Rice and Rayleigh distributions

If a species has individuals spatially distributed according to an isotropic bivariate normal distribution with standard deviation $\sigma$, and with its centre at a distance $v$ from the focal point (the origin), then the distribution of distances to the focal point follows a Rice distribution[29]. The probability density function of the Rice distribution is

$$f(r|v,\sigma) = \frac{r}{\sigma^2}\exp\left(-\frac{r^2 + v^2}{2\sigma^2}\right)I_0\left(\frac{rv}{\sigma^2}\right), \tag{4}$$

where $I_0\left(\frac{rv}{\sigma^2}\right)$ is the modified Bessel function of the first kind and order zero[33], and $\sigma$ is the scale parameter. Its cdf is

$$F(r|v,\sigma) = 1 - Q_1\left(\frac{v}{\sigma}, \frac{r}{\sigma}\right), \qquad (5)$$

where $Q_1$ is the Marcum $Q$-function[34]. The expected value of the Rice distribution is

$$v' = \sigma\sqrt{\pi/2}\,_1F_1\left(-0.5; 1; \frac{-v^2}{2\sigma^2}\right), \qquad (6)$$

where $_1F_1$ is the confluent hypergeometric function of the first kind. We calculated the $_1F_1$ function with the R package CharFun[35]. The variance is given by

$$\sigma'^2 = 2\sigma^2 + v^2 - v'^2, \qquad (7)$$

The Rayleigh distribution is a particular case of the Rice distribution when the isotropic bivariate distribution is centred at the origin. Its probability density function is

$$f(r|\sigma) = \frac{r}{\sigma^2}\exp\left(-\frac{r^2}{2\sigma^2}\right), \qquad (8)$$

where $\sigma > 0$ is the scale parameter. The cdf is

$$F(r|\sigma) = 1 - \exp\left(-\frac{r^2}{2\sigma^2}\right). \qquad (9)$$

The mean is $\sigma\sqrt{\pi/2}$ and the standard deviation $\sigma\sqrt{(4-\pi)/2}$, thus the ratio of the mean by the standard deviation is constant and equal to $\sqrt{\pi/(4-\pi)}$.

## Simulations

In the simulations, the species had isotropic bivariate normal distributions with ranges, with centres $(x_0, y_0)$ at a distance $v_p = \sqrt{x_0^2 + y_0^2}$ from the focal point. The parameter $v_p$ was uniformly randomly distributed and confined to a circle of radius $R_T$, therefore, it was sampled from $\sqrt{\text{unif}[0, R_T]}$, where 'unif' stands for 'uniform distribution'. To estimate the parameters of the GEV for a given species we sampled 1000 points (individuals) from its corresponding Rice distribution, using the R package *VGAM*[36], and identified the minimum distance. This sampling process was repeated 1000 times. The GEV parameters were then estimated from these 1000 points using the R package *extRemes*[37].

## Materials

**GBIF data**. The GBIF is the world's largest biodiversity aggregator, housing over two billion biodiversity records, a figure that has seen a 12-fold increase in available data since 2007[38]. Data retrieval from the GBIF was facilitated by the 'rgbif' package[39,40]. The data were downloaded as a .csv file (simple GBIF format) based on the taxon keys of each species. The species selection process was guided by various lists provided by Dr. Miguel Alejandro Fernandez Trigoso. Records lacking information on coordinates and with geospatial issues were excluded. The dataset encompassed all 'human observations', 'observations' and 'machine observations' across all available years. Post-download, certain records were removed due to encountering one of the following two issues: 'taxon_fuzzy_match' and 'geodetic_datum_invalid'. In addition, only records indicating an occurrence status of 'present' were retained. The observation coordinates underwent projection into equidistant projections specific for each landmass (GBIF provides data in WGS84).

Geospatial analyses were performed utilising the 'sf'[41,42] and 'terra'[43] packages and for enhanced analytical capabilities and visualisation of point data, the 'splancs' package[44] was incorporated. The home range estimations for each species were computed employing the 'adehabitatHR' package[45]. The general data processing relied on the R packages 'dplyr'[46], 'purrr'[47], and 'readr'[48], and maps were obtained with 'rnaturalearth'[49]. For each landmass (Africa, Australia, Eurasia, North America and South America) and taxonomic group (amphibians, birds, mammals and reptiles), we randomly selected the coordinates of 200 focal points. For each focal point, we determined the distance to the nearest coast, identified the species within this radius and recorded, for each species, the individual closest to the focal point. If a focal point led to fewer than 50 species, it was ignored and another focal point was chosen; this process was repeated until we reached 200 SARs. The minimum distances were then converted to the area of the corresponding circle centred on the focal point. Finally, we constructed the SAR using these areas.

To obtain the 'mean SAR', we used the following procedure. For a given number of species, say 1 species, the first observed, we ranked the SARs according to the area at each the first species was observed. For example, if we have three SARs (SAR₁, SAR₂, SAR₃), and if their first observed species is located at 1 ha, 3 ha, and 2 ha, respectively, then the SARs would be ranked (SAR₁ = 1st, SAR₂ = 3rd, SAR₃ = 2nd). We repeated this procedure for the second observed species, the 3rd, et. seq., which allowed to have for each SAR a number corresponding to its rank for a given number of species. We then calculated for each SAR the mean value of its ranks; this corresponds to a vector of the mean rank of each species. The 'mean SAR' is, then, the one whose mean rank is closer to mean of the vector of the mean ranks. We also determine the 'median SAR' as the SAR whose mean rank is closer to the median of the vector of the mean ranks. The results obtained with the 'median SAR' and the 'mean SAR' were similar.

## Reporting summary

Further information on research design is available in the Nature Portfolio Reporting Summary linked to this article.

## Data availability

We downloaded data for amphibians (A), birds (B), mammals (M), and reptiles (R) from the Global Biodiversity Information Facility for Africa (AF), Australia (AU), Asia (AS), Europe (EU), Eurasia (EA), North America (NA), and South America. The dataset is identified by the following DOI code, using the format Taxa.Landmass.Year.Month.Day, where the date corresponds to when the data was accessed: A.AF.2023.02.13 - https://doi.org/10.15468/dl.3htyck, B.AF.2023.02.13 - https://doi.org/10.15468/dl.76ykxu, M.AF.2023.02.13 - https://doi.org/10.15468/dl.94g5q8, R.AF.2023.01.30 - https://doi.org/10.15468/dl.9bnbme, A.AU.2023.02.13 - https://doi.org/10.15468/dl.52ftrc, B.AU.2023.02.13 - https://doi.org/10.15468/dl.w3r3fm, M.AU.2023.02.13 - https://doi.org/10.15468/dl.qyjkqg, R.AU.2023.01.30 - https://doi.org/10.15468/dl.u2733r, A.AS.2023.02.15 - https://doi.org/10.15468/dl.grprm8, M.AS.2023.02.15 - https://doi.org/10.15468/dl.7v9wkv, R.AS.2023.02.15 - https://doi.org/10.15468/dl.cg64ct, B.EA.2023.02.16 - https://doi.org/10.15468/dl.nmcsqs, A.EU.2023.02.15 - https://doi.org/10.15468/dl.h3xku4, M.EU.2023.02.15 - https://doi.org/10.15468/dl.dpr9jx, R.EU.2023.02.15 - https://doi.org/10.15468/dl.9r649b, A.NA.2023.02.13 - https://doi.org/10.15468/dl.uat8ch, B.NA.2023.02.14 - https://doi.org/10.15468/dl.9d4hjn, M.NA.2023.02.13 - https://doi.org/10.15468/dl.b6s9h8, R.NA.2023.01.26 - https://doi.org/10.15468/dl.3pdzsg, A.SA.2023.02.13 - https://doi.org/10.15468/dl.pwznx7, B.SA.2023.02.14 - https://doi.org/10.15468/dl.bcs9rv, M.SA.2023.02.13 - https://doi.org/10.15468/dl.ru9h67, R.SA.2023.01.30 - https://doi.org/10.15468/dl.hyjxyt. The data used in this study are available in the Zenodo database via the following link https://doi.org/10.5281/zenodo.11222083.

## Code availability

All scripts and code used to analyse the data and necessary to reproduce the figures are available from Zenodo here: https://doi.org/10.5281/zenodo.11222083.

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

## Acknowledgements

We thank Dr. Miguel Alejandro Fernandez Trigoso for supplying the list of species for the taxa used in this study. L.B.A. thanks Prof. M. Sreehari, Prof. S. Ravi and Prof. A. S. Praveena for discussions on extreme value theory. This work was supported by Norma Transitória – L57/2016/CP1440/CT0022 (L.B.A.), the Fundação para a Ciência e a Tecnologia, I.P., project UIDB/00006/2020 (CEAUL) https://doi.org/10.54499/UIDB/00006/2020 (M.M.N.) and the German Research Foundation, grant DFG FZT 118 (H.M.P. and L.Q.).

## Author contributions

L.B.A. conceived and conceptualised the study, with additional contributions from H.M.P. and M.M.N. L.B.A., L.Q. and F.S.D. collected data and performed the analyses. L.B.A., H.M.P., M.M.N. and S.P.H. discussed and interpreted the results. L.B.A. wrote the original draft, and L.Q. contributed to the Methods section. All authors reviewed and edited the final version.

## Competing interests

The authors declare no competing interests.
