## [Peer review file · Nature Communications]

Modelling the species-area relationship using extreme value theory

Corresponding Author: Dr Luís Borda-de-Água

Version 0:

Reviewer comments:

Reviewer #1

(Remarks to the Author)

This paper presents a derivation of the species area curve under a set of assumptions that basically entail a random distribution of individuals of a species within the range of the species and the random location of species ranges within a region. Conceptually, the model is very simple (e.g., the same range sizes for all species, no variation in the average density of individuals within the range, no gradients in species richness across the region of interest).

Despite these very simple assumptions, the model predicts variation in the rate of accumulation of species with increasing area. These variations in the rate of accumulation match the (admittedly roughly-estimated) patterns that are seen in real datasets. In this sense, the paper is quite elegant - a conceptually simple model generating patterns that are not immediately intuitive prior to the analysis but can be made sense of post-analysis.

I wonder if the authors could explore the degree to which the same patterns in the predictions hold if these very simple assumptions are violated. That would ideally be done via analysis, but at the least I think it deserves some consideration.

(Remarks on code availability)

I selected No for the question "Have you reviewed the code?" because I didn't review it in detail. However, I have run some of the code, I found no errors, and it seems to be comprehensive. But is the file "figure_3.dat" missing?

Reviewer #2

(Remarks to the Author)

In manuscript NCOMMS-24-42620-T, entitled "Extreme value theory explains the species-area relationship", Borda-de-Água and others employed Extreme Value Theory and computer simulations to construct a mathematical framework to describe the species-area relationship. With these simulations they describe the properties of the triphasic nested SAR and test this approach using empirical data for a variety of taxa across multiple regions around the globe. They demonstrated components of the triphasic nested SAR were evident while controlling for population abundance, employing consistent population sizes across all species. This manuscript is well-written, flows logically, and contains a considerable amount of work. However, I have several comments and suggestions I would like the authors address that I believe would improve the manuscript.

General Suggestions

As so much work has been done on the SAR, I suggest the authors outline more effectively how their approach is novel and moves the study of the SAR forward. This is impressive work, but some of the novelty of the work could be emphasized more completely. For example, it is mentioned in the Abstract that "A key insight of our study is that each phase is determined by the location of the range of species contributing to the species-area relationship relative to the focal point". For those not so immersed in the subtleties of the SAR, what are the implications of this finding? Additionally, in the Abstract, the statement is made that the "results do not hinge on the biological characteristics of the communities", yet nothing about this is mentioned in the Discussion. For these results to be informative to a more general audience, please outline the utility of your approach

and how it moves the field forward and how you envision it being implemented in future studies. In short, emphasize more the main take-aways from this work.

Also, are there limitations to your approach? Habitat heterogeneity is often considered an important determinant of the SAR. While it was obviously not an influence in the simulations, what about for the empirical datasets? Perhaps briefly address whether this might have influenced your results in the Discussion. Additionally, assumptions of normal distributions are fine for simulations, but how likely is this in real-world situations when populations of many species are often clumped? Does this potentially impact your findings if at all?

Specific Suggestions

Line 28: 'very' rather than 'vary'

Lines 32-33: consider: '...the distributions of minimum distances to a starting sampling focal point for each individual species.'

Line 38: Global Biodiversity Information Facility

Line 65: remove space between Williamson and reference

Line 76: Storch (2016, Doi: 10.1111/jvs.12428) also argued Phase III can result from many species having small ranges in comparison to sampling areas.

Lines 117-118: consider: 'the relationship between the relative position of the range for a species relative to the...'

Line 141: consider: 'the locations of individuals'

Line 163: add space before 'indicates'

Line 163: consider removing 'that'

Lines 163-164: what are the ecological implications of this?

Line 166: consider: 'suggesting the ranges for these species'

Line 178: consider: 'the ranges of each species simply as disks'

Line 195: consider: 'centre of the range'

Line 220: consider: 'the centres of the ranges for each species'

Figure 4: insert space before 'Normal like'

Line 284: 'can have'

Line 289: perhaps 'the most complete' or 'the most intensely monitored'

Line 299: 'can have'

Line 304: consider: 'to the centre of their range'

Line 309: consider: 'from the centre of the range'

Line 318: do you mean the 'green line' rather than red?

Line 342: consider: 'the ranges for each species'

Line 347: 'called'

Line 352: consider: 'the range sizes of species'

Line 353: consider: 'the starting point of the sampling scheme.'

Line 358: 'parameter'

Table S1: Is there a potential explanation for what happened with the estimates of SP(I-II) for Amphibians in Eurasia?

(Remarks on code availability)

There is a README file with enough instructions for installing and running the application. While I did not install or run any

of the code, I did look at some of the files with code and they contained many notes and descriptions that would aid in reproducing the results.

Reviewer #3

(Remarks to the Author)

What are the noteworthy results? Nice mathematical treatment (but artificial for the essence of the issue).

Will the work be of significance to the field and related fields? How does it compare to the established literature? The work raises interesting points, but with assumptions that force the conclusions. In what regards comparison with established literature, I point out that instead of the modelling used by the authors, Diggle and Allen, for instance, adopt more natural Poisson spatial patterns randomness, and other authors lognormal. Their restriction to normal / Rice / Rayleigh provides nice mathematical treatment and a relationship to EVT, but seems arbitrary.

Does the work support the conclusions and claims, or is additional evidence needed?

Are there any flaws in the data analysis, interpretation and conclusions? The data analysis and mathematical derivations are fine, but starting with arbitrary assumptions, such as uniform distribution of focal points or Rayleigh / Ride / normal scattering patterns, and therefore conclusions are doubtful.

Is the methodology sound? In my view there are many unproved assumptions (or even inadequate, see that in line 930 the fitness of a normal distances model is rejected) leading to unproved claims, namely that EVT explains SARS.

Is there enough detail provided in the methods for the work to be reproduced? Yes in what regards simulation, doubtful in what regards the use of real data.

Report:

Report on Extreme value theory explains the species-area relationship

by Luís Borda-de-Agua, M. Manuela Neves, Luise Quoss, Stephen P. Hubbell, Filipe S. Dias, Henrique M. Pereira

Abstract

line 28: "it increases vary rapidly" it increases very rapidly ?

lines 31-33: "and show that it is the mixture of the distributions of each species individuals' minimum distances to a starting sampling focal point" "it" refers to what? Surely not to "theory", or "relationship", or "EVT", since none of them can be considered a mixture of distributions.

Line 34: "location of the range". At this stage what is the location of the range is not clear.

Lines 61-64: "Phase I, the number of species increases rapidly, with a slope approaching 1. Subsequently, for intermediate area sizes, Phase II, the SAR exhibits slower growth, following a power law with an exponent smaller than one. Finally, for large area sizes, Phase III, the SAR experiences rapid growth again, with a slope tending to one." This is obvious, since when a species has been found it is counted and afterwards disregarded. What I would like to understand is the claim about the slopes values, since I think that they must depend on the focal point. As in the last sentence in the abstract the authors say that their results "can be widely applicable to systems that exhibit similar spatial features" I did a rough experiment counting surnames of habitation owners and tenants of habitations in the 32 London boroughs.

As expected, results with the focal point in the middle of 34 were different from results with focal point at the frontier of boroughs 6,7,19,20,21, namely because immigration introduced non traditional surnames. In what concerns animal species, something similar must occur with location of resources, as water (which attracts many species, thus providing food to predators, and is the main cause for massive migrations), and moreover there is ample evidence that the rank-size constraints originate Zipf-Mandelbrot counts (this may be the reason of

Lines 67-68: "In the limiting case where species are represented by only one individual" this is expected in the Zipf theory.

Line 69: "the power law relationship". Beware of the temptation of power laws, a caveat implied in papers such as Stumpf, M. P. H., and Porter, M. A. (2012). Critical Truths about Power Laws. *Science*, 335(6069), 665–666, in Shalizi, C. R. (2007). So You Think You Have a Power Law? Well Isn't That Special? bactra.org/weblog/491.html or The Econophysics Blog (2006). Tyranny of the Power Law (and Why We Should Become Eclectic). <http://econophysics.blogspot.com/2006/07/tyranny-of-power-law-and-why-we-should.html>

Lines 70-71: "assuming an underlying lognormal species abundance distribution" The use of lognormal models stems out from the fact that it mimics the linear signature of power laws in log-log plots, a feature explained in the pioneer paper Belevitch, V. (1959). On the Statistical Laws of Linguistic Distributions. *Annales de la Société Scientifique de Bruxelles, Ser. I*, 73, 310–326. <http://www.csl.sri.com/users/neumann/belevitch.pdf>. Once again, confounding simple serendipity with the essence of the question.

Line 102: "can be defined as an equally weighted sum". I don't see a justification to consider "equally".

Line 111-112: "... Gumbel distributions" A comment on the many references (and I didn't check most of them): what is the reason for so many references when a single one would be enough? For instance in what regards EVT, aside from the books 7, 8, 9, 26, at the end of the supplementary materials the initial papers by Fréchet (48) and Fisher and Tippett (49), and also a paper by Gumbel (50) are listed. Speaking of references, for instance in references 49 and 50 is used a "in" whereas the publication is a journal, not an edited book. Another thing on the references list, 17, 29 and 38 use as authors [first author] et al., which I consider inappropriate. I also consider that references seem exaggerated, what is for instance the relevance of a package (34) for transforming characteristic functions (that are relevant for sums but not for extremes) in CDF's or PDF's?

Lines 124-125: "distributed according to bivariate normal distributions with scale parameter $p = 1$ and randomly uniform centres." Those arbitrary assumptions will unduly narrow down the EVT models to Weibull-0.5 and to Gumbel (although the slow and regularly varying left tail variation characterization of domains of attraction of stable EVT laws are never referred).

Observe that in Lines 927- 931 the goodness of fit of the normal model is rejected.

Line 141: "cases an discuss" cases and discuss

Lines 154-155: "assuming the individuals' locations to be isotropic bivariate Cauchy distributions". The Cauchy is additive stable with index 1, so due to the similarity of characterizations of the domains of attraction of stable laws in the additive and in the EVT contexts it is in the EVT domain of attraction of the Fréchet 1, so the claim that the "results were qualitatively similar" is vague.

Line 199: what is the meaning of $\xi \leq 0$?

Lines 222 and 226: $sP(I-II) = sP(I-II)/ST$, $sP(II-III) = sP(II-III) /ST$ ST, the minima of all species (line 103) is 1?

Line 276: "We only considered SARs that had at least 50 species". Arbitrary assumptions can provide interesting "findings", frequently spurious and arbitrary. The assumptions made in the simulations section, and decisions as this one in line 276, will indeed influence the conclusions, and I don't think that the title and main claim that EVT explains SAR is valid, the relationship results from the arbitrary assumption that the nearest neighbor of the focal point has normal, Rice or Rayleigh distribution. Observe that for instance Diggle, P.J. 1983. Statistical Analysis of Spatial Point Patterns. New York: Academic Press recommends spatial Poisson randomness, used by Allen and White (reference 51), and many researchers in the field use lognormal (multiplicative pattern, but conforming with power laws signature) instead of normal (additive pattern). All this of course provides a huge variety of regular variation, and establishing connections with EVT seems more a serendipity issue than an essential finding. Arbitrary assumptions and restrictions have the flavor of the ironical guidance to postgraduate students "get the data that fit your solution".

Line 293: "For each taxon and landmass, we obtained 200 SARs" for five landmasses — why there is no indication of the radii of the phases disks?

Figure 5: I detected daltonism, since I see a green line and the caption speaks of a red line indicating a mean SAR. Why no confidence bands, the diversity of the grey lines showing a widespread of variability?

Lines 327-329: "Here, we showed that extreme value theory offers a comprehensive framework for analysing the SAR, providing a unified explanation for the characteristics of the three phases" As said before, I don't agree with this bold claim, and more generally I don't think the main conclusions are properly supported.

Methods

The Rice and Rayleigh distributions, OK, nothing new but collects information that many readers are unfamiliar with. As said before, some references, such as 34, seem forced.

Simulations and Materials The comments on the main body of the paper indicate our reserves on this.

Lines 491-492: "1ha, 3ha, and 2ha". This gives an idea of the length of the radii used, but the absence of that indication is strange. In line 492, it should be 2nd instead of 2rd.

The exact and asymptotic distributions of the minima Ok, but a minimal indication on the characterization of the domains of attraction of EVT stable laws would provide a better focus on the issues.

The Disk Model Aside the comment on radii, I repeat the comment that in nature resources and competition are two main causes for species dispersion. For instance, along a river I doubt that the disk model is adequate, whereas it can be useful around an isolated lagoon. On the other hand, it is known that under or overdispersion indicated by the dispersion index $\text{Var}(X)/E(X)$ is an important parameter, the Poisson distribution with $\text{Var}(XP)/E(XP)=1$ modelling unrestricted randomness (and related to maximum entropy) very much praised, eventually because with only one parameter it is an example of parsimony in fitting (aside from the traditional counting models related to Bernoulli trials, Fisher used a logartmc model, that depending on the parameters may be over or underdispersed, and Engen developed an extended negative binomial model for natural populations).

Although this paper deals appropriately with sophisticated mathematical issues, those seem more a way of jumping to conclusions resulting from unproved assumptions than a real explanation of Nature patterns. So, recognizing the mathematical merits but doubting of their appropriateness to provide new insights in the field, and moreover not accepting the main claim that EVT explains SAR, I don't recommend, in the present form, the acceptance of this manuscript (and it is not a question of minor revision, the questions I raise are in the essence of the conception).

(Remarks on code availability)

Version 1:

Reviewer comments:

Reviewer #2

(Remarks to the Author)

I have had an opportunity to carefully review the revised manuscript NCOMMS-24-42620A, Modelling the species-area relationship using extreme value theory.

It is my opinion the authors have done a nice job of incorporating my comments and those from the other reviewers. I feel that the revisions have greatly improved the readability and flow of the manuscript. Specifically, the authors have better emphasized the novelty of their findings and the implications of their work for the broader field of ecology. I also believe they have done a nice job describing the limitations of their approach and the added simulations to address a wider range of scenarios have greatly improved the work. I believe this work makes an important contribution to the field of ecology and will be of interest to the readers of *Nature Communications*.

(Remarks on code availability)

While I did not install and run all of the code, all of the scripts contained many notes and descriptions that would aid in reproducing the results. The readme file also contained enough instructions for running the applications.

Reviewer #3

(Remarks to the Author)

The revised version has taken into account almost all the observations I had made in the first round of the evaluation.

Namely, the authors have changed the title of their paper, the former one being an overstated claim.

In their rebuttal, the authors write that the Benford-Zipf-Mandelbrot "counts seem hardly of relevance for our analysis". I disagree, the rank-size equilibrium predicts a large number of species with a tiny number of individuals, that therefore can escape scrutiny.

However this is a minor point, since Springer states that referee's comments and authors rebuttal will be released alongside with accepted papers. Hence, interested readers can make their own judgement.

Therefore, I have no longer reservations on the acceptance of this submission.

(Remarks on code availability)

REVIEWER COMMENTS

The reviewers' original comments are in bold, and our replies are in regular text.

Reviewer #1 (Remarks to the Author):

This paper presents a derivation of the species area curve under a set of assumptions that basically entail a random distribution of individuals of a species within the range of the species and the random location of species ranges within a region. Conceptually, the model is very simple (e.g., the same range sizes for all species, no variation in the average density of individuals within the range, no gradients in species richness across the region of interest).

Despite these very simple assumptions, the model predicts variation in the rate of accumulation of species with increasing area. These variations in the rate of accumulation match the (admittedly roughly-estimated) patterns that are seen in real datasets. In this sense, the paper is quite elegant - a conceptually simple model generating patterns that are not immediately intuitive prior to the analysis but can be made sense of post-analysis.

I wonder if the authors could explore the degree to which the same patterns in the predictions hold if these very simple assumptions are violated. That would ideally be done via analysis, but at the least I think it deserves some consideration.

We thank the reviewer for their careful reading of our manuscript. In response to the comment regarding the last paragraph, we have significantly expanded Section D (Simulations) of the Supplementary Information to address a wider range of scenarios. Although the conclusions of the main text do not change, the new simulations provide additional insights into some of the SARs observed with the GBIF data. Although we agree these are important results (and probably should have been included in the initial version of the manuscript), we prefer to keep them in the Supplementary Information to maintain a clear and concise message in the main text. We added the following sentences to the main text:

“In addition to the previous simulations, we also performed further simulations across a wider range of scenarios, including (i) species with varying range sizes, (ii) spatially non-uniformly distributed ranges and (iii) species with different number of individuals and range sizes; the results are presented in Supplementary Information D, Fig. S3-S10. While the main attributes of the SARs and their interpretation through the EVT do not change, these simulations provide further insights into the patterns observed in empirically observed SARs (see Supplementary Information F).”

Reviewer #1 (Remarks on code availability):

I selected No for the question "Have you reviewed the code?" because I didn't review it in detail. However, I have run some of the code, I found no errors, and it seems to be comprehensive. But is the file "figure_3.dat" missing?

Thank you very much for bringing this to our attention. It was indeed an oversight on our part. We have added the file “Data_Figures_2_3_S2_S3_S4.zip” to Zenodo, which contains the data required for the following functions: figure_2.R, figure_3.R, figure_S2.R, figure_S3.R and figure_S4.R.

Reviewer #2 (Remarks to the Author):

In manuscript NCOMMS-24-42620-T, entitled “Extreme value theory explains the species-area relationship”, Borda-de-Água and others employed Extreme Value Theory and computer simulations to construct a mathematical framework to describe the species-area relationship. With these simulations they describe the properties of the triphasic nested SAR and test this approach using empirical data for a variety of taxa across multiple regions around the globe. They demonstrated components of the triphasic nested SAR were evident while controlling for population abundance, employing consistent population sizes across all species. This manuscript is well-written, flows logically, and contains a considerable amount of work. However, I have several comments and suggestions I would like the authors address that I believe would improve the manuscript.

We thank the reviewer for their careful reading of our manuscript and their suggestions to improve our work.

General Suggestions

As so much work has been done on the SAR, I suggest the authors outline more effectively how their approach is novel and moves the study of the SAR forward. This is impressive work, but *some of the novelty of the work could be emphasized more completely*. For example, it is mentioned in the Abstract that “A key insight of our study is that each phase is determined by the location of the range of species contributing to the species-area relationship relative to the focal point”. For those not so immersed in the subtleties of the SAR, what are the implications of this finding? Additionally, in the Abstract, the statement is made that the “results do not hinge on the biological characteristics of the communities”, yet nothing about this is mentioned in the Discussion. For these results to be informative to a more general audience, *please outline the utility of your approach and how it moves the field forward and how you envision it being implemented in future studies. In short, emphasize more the main take-aways from this work.*

Thank you for this important comment. We highlighted in italic what we believe to be its most important messages, and accordingly, we have changed the first paragraph of the discussion which now reads:

“Although the SAR is a basic pattern with a long and venerable history in ecological studies¹, it remains an active area of research²⁴. Here, we show that extreme value theory offers a comprehensive framework for analysing the SAR. The power of the EVT lies in its ability to provide several analytical tools and conceptual insights for interpreting the characteristics of the SAR at different spatial scales. For instance, under certain idealized conditions, the steep slopes observed for small spatial scales (Phase I) and large spatial scales (Phase III) can be derived and interpreted using EVT. Our work is a first incursion into the relationship between EVT and an ecological pattern, the SAR, and we anticipate that future applications of EVT, and order statistics, in ecology will uncover new patterns and provide tools for their quantification, potentially linking these patterns to their underlying processes. Probably, one of the most significant insights from our work is the discovered relationship between a species range and location and the SAR phase to which it contributes. This observation led to

the development of a simple rule of thumb for predicting the number of species at which transitions among the phases of the SAR occur ($S_{P(I-II)}$ and $S_{P(II-III)}$); see also Supplementary Information E. This rule may be further refined as more detailed information on species ranges is collected. We foresee that $S_{P(I-II)}$ and $S_{P(II-III)}$ may be used in conservation studies aimed at estimating species richness and identifying the areas sizes where changes in the rate at which new species appear are expected.”

- Regarding the sentence “**A key insight of our study is that each phase is determined by the location of the range of species contributing to the species-area relationship relative to the focal point**” we changed it slightly and merged it with the next one. We believe this revision more effectively highlights the implications of our finding:

“A key insight of our study is that each phase is determined by the geographical distributions of the species, i.e., their ranges, relative to the focal point, which in turn led us to develop a formula for estimating the number of species at the transition between the phases.”

- Regarding the sentence “**results do not hinge on the biological characteristics of the communities**” we have made a slight revision, because we believe the new version provides a more accurate interpretation of our original intent:

“Although a SAR reflects the underlying biological attributes of the constituent species (e.g., dispersal ability), our interpretations and use of the EVT are general and can be widely applicable to systems that exhibit similar spatial features.”

We revisit this topic in the last paragraph of the Discussion, where we state:

“That is, the functional form of the nested SAR is not idiosyncratic to ecological communities but rather arises of general “laws acting around us”, borrowing an expression by J. Harte³². It is the numerical values of a specific SAR parameter that reflect the underlying biology of the species and the interactions among themselves and with the environment. This suggests that the findings of our work can be applied to any systems where the components form spatial aggregates similar to those observed among species in communities.”

Also, are there limitations to your approach? Habitat heterogeneity is often considered an important determinant of the SAR. While it was obviously not an influence in the simulations, what about for the empirical datasets? Perhaps briefly address whether this might have influenced your results in the Discussion.

Reviewer #1 had a very similar comment, and we acknowledge that some results should have been presented in the initial version of the manuscript. We have significantly expanded Section D (Simulations) of the Supplementary Information to address a wider range of scenarios. Although the main conclusions of the main text do not change, the new simulations provide additional insights into some of the SARs observed with the GBIF data. However, we prefer to keep these results in the Supplementary Information to maintain a clear and concise message in the main text. We added the following sentences to the main text:

“In addition to the previous simulations, we also performed further simulations across a wider range of scenarios, including (i) species with varying range sizes, (ii) spatially non-uniformly distributed ranges and (iii) species with different number of individuals and range

sizes; the results are presented in Supplementary Information D, Fig. S3-S10. While the main attributes of the SARs and their interpretation through the EVT do not change, these simulations provide further insights into the patterns observed in empirically observed SARs (see Supplementary Information F).”

Additionally, assumptions of normal distributions are fine for simulations, but how likely is this in real-world situations when populations of many species are often clumped? Does this potentially impact your findings if at all?

It is important to note that an isotropic bivariate normal distribution of the individuals of a given species represent a clumped distribution of points. However, we acknowledge that real communities often exhibit a gradient in species richness, that is, the centres of the distributions are not uniformly randomly distributed. To address this, we now present simulations that incorporate such gradients in species richness in Supplementary Information D. These new results show that the conclusions in the main text remain unchanged. For the reasons previously stated, we prefer to keep these results in the Supplementary Information.

Specific Suggestions

Line 28: ‘very’ rather than ‘vary’

We thank the reviewer for pointing out this and other typos in the original manuscript. We changed accordingly.

Lines 32-33: consider: ‘...the distributions of minimum distances to a starting sampling focal point for each individual species.’

We changed as recommended.

Line 38: Global Biodiversity Information Facility

We changed as recommended.

Line 65: remove space between Williamson and reference

We removed the space.

Line 76: Storch (2016, Doi: 10.1111/jvs.12428) also argued Phase III can result from many species having small ranges in comparison to sampling areas.

We agree with this statement and thank the reviewer to bringing this paper to our attention. We added this reference to the line mentioned (as well as to the reference list).

Lines 117-118: consider: ‘the relationship between the relative position of the range for a species relative to the...’

We changed the sentence as recommended.

Line 141: consider: 'the locations of individuals'

We changed as recommended.

Line 163: add space before 'indicates'

We added the space.

Line 163: consider removing 'that'

We removed "that".

Lines 163-164: what are the ecological implications of this?

We would not ascribe much ecological significance to the statements in these lines. They primarily concern the relative position of the centre of a distribution and its range to the focal point, and how this influences the phase of the SAR to which it is most likely to contribute. Therefore, the relationship described is largely geometrical, reflecting the spatial arrangements of the species in relation to the focal point.

Line 166: consider: 'suggesting the ranges for these species'

We changed the sentence as recommended.

Line 178: consider: 'the ranges of each species simply as disks'

We changed the sentence as recommended.

Line 195: consider: 'centre of the range'

We changed the sentence as recommended.

Line 220: consider: 'the centres of the ranges for each species'

We changed the sentence as recommended.

Figure 4: insert space before 'Normal like'

We inserted the space before "Normal like"

Line 284: ‘can have’

We changed the sentence as recommended.

Line 289: perhaps 'the most complete' or 'the most intensely monitored'

We adopted the expression “the most complete”.

Line 299: ‘can have’

We changed as recommended.

Line 304: consider: 'to the centre of their range'

We changed as recommended.

Line 309: consider: 'from the centre of the range'

We changed as recommended.

Line 318: do you mean the ‘green line’ rather than red?

Thank you for pointing that mistake. We indeed meant the “green line”, not the “red line”, and we have corrected this error accordingly.

Line 342: consider: 'the ranges for each species'

We changed as recommended.

Line 347: ‘called’

We changed “call” to “called”.

Line 352: consider: 'the range sizes of species'

We changed as recommended.

Line 353: consider: ‘the starting point of the sampling scheme.’

We changed as recommended.

Line 358: ‘parameter’

We changed “parameters” to “parameter”

Table S1: Is there a potential explanation for what happened with the estimates of SP(I-

II) for Amphibians in Eurasia?

There is indeed an explanation for these estimates. We have included an example of a simulation in Supplementary Information D that produces an outcome similar to that observed for amphibians in Eurasia (Fig. S7 and S8). In brief, this situation occurs when the species ranges (or only a very few) include the focal point.

Reviewer #2 (Remarks on code availability):

There is a README file with enough instructions for installing and running the application. While I did not install or run any of the code, I did look at some of the files with code and they contained many notes and descriptions that would aid in reproducing the results.

Reviewer #3 (Remarks to the Author):

What are the noteworthy results? Nice mathematical treatment (but artificial for the essence of the issue).

Will the work be of significance to the field and related fields?

How does it compare to the established literature? The work raises interesting points, but with assumptions that force the conclusions.

Thanks for the compliment on the analysis. We do think the mathematical treatment is fit for the analysis of the triphasic SAR and our assumptions are made to make the problem tractable mathematically. We have used computation simulations to see what happens where the assumptions are relaxed, and our theory is robust to weakening of our assumptions.

In what regards comparison with established literature, I point out that instead of the modelling used by the authors, Diggle and Allen, for instance, adopt more natural Poisson spatial patterns randomness, and other authors lognormal.

The reference to Diggle probably corresponds to the book “Diggle, P.J. 1983. Statistical Analysis of Spatial Point Patterns” (mentioned later in the report). This book covers several types of spatial point patterns, from which the Poisson point process is just one. In fact, the Poisson process corresponds to complete spatial randomness, and we can hardly say it is a “more natural” pattern (it is used as a null model to assess whether natural distributions are aggregated, regular, or indeed a Poisson point process when using the Ripley’s function). In his book, Diggle describes the Neyman-Scott model, which is very close to our simulations, as we explain in our response to the comments “Line 276”. Therefore, claiming that Diggle adopted “Poisson spatial patterns randomness” as a more “natural” approach seems to be unjustified.

The reference to Allen in this sentence (as we later found in the report) is to the paper by Allen and White (2003):

Allen, A. P. & White, E. P. Effects of range size on species-area relationships. *Evol. Ecol. Res.* **5**, 493–499 (2003).

In their paper, Allen and White do not address the distribution of individuals of a given species. Instead, they deal with the spatial distribution of ranges, exactly like we did in our “disk model” (we show in the manuscript that the two approaches are equivalent). Incidentally, in the simulations presented in the main text, we assumed that the centres of the ranges to be uniformly randomly distributed, corresponding to a Poisson point process.

Lastly, the reference to the lognormal distribution is ambiguous. It is unclear what the lognormal distribution is supposed to be applied to: the spatial distribution of points, the distribution of the distances of points to the focal point, or something else? However, the lognormal distribution is often used to describe species abundance distributions, which describes in a community how many species have a given number of individuals. This is not the same as the spatial distribution of the individuals within a species. Is it possible that Reviewer #3 is mentioning the lognormal in the context of species abundance distributions and therefore mixing two entirely different concepts? (A suspicion that arose while reading their comment on “Lines 70-71”.)

Their restriction to normal / Rice / Rayleigh provides nice mathematical treatment and a relationship to EVT, but seems arbitrary.

We do not restrict to Normal/Rice/Rayleigh in general. Our only assumption is that the distribution of points of a species follows a normal distribution, an assumption often used in other studies of the SAR and species distributions patterns (Plotkin et al. 2000).

Plotkin, J. B., Potts, M. D., Leslie, N., Manokaran, N., LaFrankie, J., & Ashton, P. S. (2000). Species-area curves, spatial aggregation, and habitat specialization in tropical forests. *Journal of theoretical biology*, *207*(1), 81-99.

Does the work support the conclusions and claims, or is additional evidence needed? Are there any flaws in the data analysis, interpretation and conclusions? The data analysis and mathematical derivations are fine, but starting with arbitrary assumptions, such as uniform distribution of focal points or Rayleigh / Rice / normal scattering patterns, and therefore conclusions are doubtful.

We appreciate the reviewer acknowledgement that our analyses and mathematical derivations are correct. Regarding the simulations, there is only one focal point, and we assumed that the centres of the distributions were uniformly, randomly distributed and that the distributions of individuals follow an isotropic bivariate normal distribution. These assumptions were made for two reasons: (1) they provide a very simple model, and (2) they are the basic tenets of the Neyman-Scott model (e.g. Cressie 1993, Diggle 2013). We elaborate further on these points in our response to the comment concerning Line 276. The Rayleigh and the Rice distributions are a direct consequence of assuming an isotropic bivariate normal distribution.

In this context, the suggestion that the “conclusions are doubtful” seems misplaced. It is important to emphasize that the purpose of using this model in the simulations was to demonstrate that even such a model with a simple set of assumption can reproduce the “classical” triphasic SAR. The results obtained using the GBIF data, however, are entirely independent of these assumptions. They serve to illustrate the variety of SARs observed in real-world data and that some of the findings from the simulations can be applied to empirical datasets.

Cressie, N. (1993). *Statistics for spatial data*. John Wiley & Sons.

Diggle, P. J. (2013). *Statistical analysis of spatial and spatio-temporal point patterns*. CRC press.

Is the methodology sound? In my view there are many unproved assumptions (or even inadequate, see that in line 930 the fitness of a normal distances model is rejected) leading to unproved claims, namely that EVT explains SARS.

See below.

Is there enough detail provided in the methods for the work to be reproduced? Yes in what regards simulation, doubtful in what regards the use of real data.

Not clear what the reviewer mentions here by “doubtful” in what regards the use of real data. We use both simulations and real data, both of them available in our code.

Report:

Report on Extreme value theory explains the species-area relationship

by Luís Borda-de-Água, M. Manuela Neves, Luise Quoss, Stephen P. Hubbell, Filipe S. Dias, Henrique M. Pereira

Abstract

line 28: “it increases vary rapidly” it increases very rapidly ?

Thank you for pointing out this mistake. We have changed accordingly.

lines 31-33: “and show that it is the mixture of the distributions of each species individuals’ minimum distances to a starting sampling focal point” “it” refers to what? Surely not to “theory”, or “relationship”, or “EVT”, since none f them can be considered a mixture of distributions.

Thank you for bringing this to our attention. We have changed the sentence to “Here we develop a theory for the SAR using Extreme Value Theory⁷⁻⁹, and show that the SAR is the mixture of...”

Line 34: “location of the range”. At this stage what is the location of the range is not clear.

Thank you for bringing this to our attention. The sentence in question has been revised to read:

“A key insight of our study is that each phase is determined by the geographical distributions of the species, i.e., their ranges, relative to the focal point.”

Lines 61-64: “Phase I, the number of species increases rapidly, with a slope approaching 1. Subsequently, for intermediate area sizes, Phase II, the SAR exhibits slower growth, following a power law with an exponent smaller than one. Finally, for large area sizes, Phase III, the SAR experiences rapid growth again, with a slope tending to one¹⁸.” This is obvious, since when a species has been found it is counted and afterwards disregarded. What I would like to understand is the claim about the slopes values, since I think that they must depend on the focal point.

The question of the slopes attained in the several phases was a major focus of our work. In Supplementary information, Appendix A (and, in particular, section *The asymptotic distribution of the minima of the Rayleigh distribution*) addresses the slope of Phase I, by relating it to the characteristics of the minimum of the Rayleigh distribution, Section B (“The Disk Model”) deals with the slope of Phase III using a simple model that we called “the disk model”, and Appendix C (“The SAR and the evolution of parameters of the generalized extreme value (GEV) distribution”) links the slope of Phase II to other characteristics of the SAR. Admittedly, these are simple models, but we believe they offer an initial understanding of the different slope magnitudes observed in real SARs.

As in the last sentence in the abstract the authors say that their results “can be widely applicable to systems that exhibit similar spatial features” I did a rough experiment counting surnames of habitation owners and tenants of habitations in the 32 London boroughs.

As expected, results with the focal point in the middle of 34 were different from results with focal point at the frontier of boroughs 6,7,19,20,21, namely because immigration introduced non traditional surnames. In what concerns animal species, something similar must occur with location of resources, as water (which attracts many species, thus providing food to predators, and is the main cause for massive migrations), and moreover there is ample evidence that the rank-size constraints originate Zipf-Mandelbrot counts (this may be the reason of Lines 67-68: “In the limiting case where species are represented by only one individual” this is expected in the Zipf theory.

Unfortunately, we are not provided details of the results obtained with the “surnames” including what was the concept of distance used so we cannot really respond to this comment. The Zipf_Mandelbrot counts seem hardly of relevance for our analysis.

Line 69: “the power law relationship”. Beware of the temptation of power laws, a caveat implied in papers such as Stumpf, M. P. H., and Porter, M. A. (2012). Critical Truths about Power Laws. *Science*, 335(6069), 665–666, in Shalizi, C. R. (2007). So You Think You Have a Power Law? Well Isn’t That Special? bactra.org/weblog/491.html or The Econophysics Blog (2006). Tyranny of the Power Law (and Why We Should Be- come Eclectic). <http://econophysics.blogspot.com/2006/07/tyranny-of-power-law-and-why-we-should.html>

We thank the reviewer for this cautionary remark. However, we would point out that the power law relationship is well documented for the Phase II, providing, at least as a first approximation) a good fit, as reviewed in length by

Rosenzweig, M. L. *Species Diversity in Space and Time*. *Species Diversity in Space and Time* (Cambridge University Press, 1995)

arises in several theories, e.g.:

Hubbell, S. P. *The Unified Neutral Theory of Biodiversity and Biogeography*. (Princeton University Press, 2001),

O'Dwyer, J. P., & Green, J. L. Field theory for biogeography: a spatially explicit model for predicting patterns of biodiversity. *Ecol. Lett.* **13**, 87-95. (2007),

and computer simulations:

Rosindell, J. & Cornell, S. J. Species-area relationships from a spatially explicit neutral model in an infinite landscape. *Ecol. Lett.* **10**, 586–595 (2007).

Lines 70-71: “assuming an underlying lognormal species abundance distribution” The use of lognormal models stems out from the fact that it mimics the linear signature of power laws in log-log plots, a feature explained in the pioneer paper Belevitch, V. (1959). *On the Statistical Laws of Linguistic Distributions*. *Annales de la Société Scientifique de Bruxelles, Ser. I*, **73**, 310–326. <http://www.csl.sri.com/users/neumann/belevitch.pdf>. **Once again, confounding simple serendipity with the essence of the question.**

We thank the reviewer to call our attention to Belevitch’s work. However, we don’t see its relevance in the context where we mentioned lognormal distributions.

This comment refers to the following sentence:

“Preston¹¹ explained the power law relationship, with a presumed exponent approximately equal to 0.25, by assuming an underlying lognormal species abundance distribution...”.

This assertion can be easily verified in the original paper, as well as in May (1975) or more recent works, such as, Hubbell (2001, p. 34). It should be noted that Preston was deriving the exponent of a power law describing a *species-area relationship*, assuming (among other things) a lognormal distribution for the *species abundance distribution*, that is, the distribution describing the number of species with a given number of individuals.

Line 102: “can be defined as an equally weighted sum”. I don’t see a justification to consider “equally”.

The cdf of a mixture distribution of cdfs $P_i(x)$ is

$$F(x) = \sum_{i=1}^n w_i P_i(x)$$

where w_i are weights, such as, $\sum_{i=1}^n w_i = 1$

(e.g. https://en.wikipedia.org/wiki/Mixture_distribution).

The mixture $S_M(A)$, defined in line 104 (and note the new notation introduced to improve readability and better align with common practices in probability theory texts), can also be written as

$$S_M(A) = \frac{1}{S_T} \sum_{i=1}^{S_T} L_i(A) = \sum_{i=1}^{S_T} \frac{1}{S_T} L_i(A).$$

In the latter formula, $1/S_T$ can be identified as the w_i in the above formula, thus all the weights are equal to $1/S_T$ and, obviously, $\sum_{i=1}^{S_T} 1/S_T = 1$.

In order to make this point clear in the manuscript, we changed the sentence in lines 102 and 103 to

“ $S_M(A)$, can be defined as an equally weighted sum, with weights equal to $1/S_T$, of the distributions of the minima of all species, S_T ..”

Line 111-112: “... Gumbel distributions7–9 A comment on the many references (and I didn’t check most of them): what is the reason for so many references when a single one would be enough? For instance in what regards EVT, aside from the books 7, 8, 9,26, at the end of the supplementary materials the initial papers by Fréchet (48) and Fisher and Tippet (49), and also a paper by Gumbel (50) are listed.

Most ecologists are not familiar with extreme value theory, therefore, we wanted to ensure that several key references - those we consider essential - were mentioned. We regularly use

7. Galambos, J. *The Asymptotic Theory of Extreme Order Statistics, Second Edition*. (Robert E. Krieger Publishing Company, 1987).
8. Coles, S. *An Introduction to Statistical Modeling of Extreme Values*. (Springer, 2001).
9. Castillo, E., Hadi, A. S., Balakrishnan, N. & Sarabia, J. M. *Extreme Value and Related Models with Applications in Engineering and Science*. (Wiley, 2005).

We consider these three references to be complementary, each providing different insights and methods. This is especially true for references 8 and 9, while reference 7 provides a more in-depth analysis of EVT. Reference 26 is not on EVT but on order statistics, a similar but distinct subject from EVT.

Again, since EVT is not widely known among ecologists, we have included three seminal works in the Supplementary Material for interested readers, references 48-50.

Speaking of references, for instance in references 49 and 50 is used a “in” whereas the publication is a journal, not an edited book.

Thank you for pointing out this mistake. We have corrected these references.

Another thing on the references list, 17, 29 and 38 use as authors [first author] et al., which I consider inappropriate.

We are following the Journal’s rules, which state that:

“All authors should be included in reference lists unless there are six or more, in which case only the first author should be given, followed by 'et al.'”

<https://www.nature.com/ncomms/submit/how-to-submit#:~:text=section%20upon%20publication,-.References,is%20given%20for%20each%20number.>

I also consider that references seem exaggerated, what is for instance the relevance of a package (34) for transforming characteristic functions (that are relevant for sums but not for extremes) in CDF's r PDF's?

The package to which reference 34 refers to, was used to estimate the confluent hypergeometric function of the first kind, as mentioned in Methods (though the name of package is admittedly uninformative in this respect). We believe it is important for readers to have the reference to the package used.

Lines 124-125: “distributed according to bivariate normal distributions with scale parameter $p=1$ and randomly uniform centres.” Those arbitrary assumptions will unduly narrow down the EVT models to Weibull-0.5 and to Gumbel (although the slow and regularly varying left tail variation characterization of domains of attraction of stable EVT laws are never referred).

Not all spatial distributions lead to a Weibull distribution with $\xi=-0.5$ or a Gumbel distribution ($\xi=0$). This only occurs for species whose range centres are either very close to the focal point or very far apart, as illustrated in Fig. 3c; in this figure we can see that in the region corresponding to Phase II, $-0.5 \leq \xi \leq 0$. Naturally, the Weibull and Gumbel distributions are the two possible limiting distributions of extremes when $\xi \leq 0$, the opposite would imply a parent distribution with a left heavy tail. The latter scenario was explored in the Supplementary Information, Fig. S11, where individuals were spatially distributed according to an isotropic bivariate Cauchy distribution. In this case, we do observe $\xi=1$ for Phase III (indicating a Fréchet distribution), as expected, as the parent distribution has a heavy tail, which in turn leads to a Fréchet distribution for the minima. As mentioned in the manuscript, real-world scenarios likely contain the both types of parent distributions.

Observe that in Lines 927- 931 the goodness of fit of the normal model is rejected.

It should be noted that the plots referred to lines 927-931 represent the distribution of the range sizes while our choice of a bivariate normal distribution was for the distribution of the individuals of one species. Therefore, there is no connection between these plots and our assumption of bivariate normal distribution for the distribution of individuals. This assumption has been explained in another response and it is based mainly on the Neyman-Scott model.

Line 141: “cases an discuss” cases and discuss

Thank you for pointing out this mistake (we believe the reviewer refers to line 242). We have changed accordingly.

Lines 154-155: “assuming the individuals' locations to be isotropic bivariate Cauchy

distributions”. The Cauchy is additive stable with index 1, so due to the similarity of characterizations of the domains of attraction of stable laws in the additive and in the EVT contexts it is in the EVT domain of attraction of the Fréchet 1, so the claim that the “results were qualitatively similar” is vague.

We have expanded this sentence:

“We also performed simulations assuming the individuals’ locations to be isotropic bivariate Cauchy distributions. The results were qualitatively similar: we observed three phases, with Phases I and III characterized by slopes close to 1 in a log-log plot, and Phase II well approximated by a power law with a slope smaller than 1, but extending over a wider range of areas compared to the simulations using isotropic bivariate normal distributions; see Supplementary Information D and Fig. S3.”

Line 199: what is the meaning of $\xi \lesssim 0$?

In order to make clear the reason why we mention that $\hat{\xi}$ is negative but approaches zero, we change the sentence where this is mentioned. It now reads:

“On the other hand, when $v_p/\sigma_p > 4$, then $\hat{\xi}$ is still negative but approaches zero ($\hat{\xi} \lesssim 0$) which is characteristic of a distribution of minima following a Gumbel distribution. This behaviour is expected if the distribution of distances follows, or approximates, a normal distribution (see Fig. 4.)”

Lines 222 and 226: $sP(I-II) = sP(I-II)/ST$, $sP(II-III) = sP(II-III) /ST$ ST, the minima of all species (line 103) is 1?

We thank the reviewer for raising this point. We believe this comment arose because the distinction between the uppercase “S” and lowercase “s” in the formulas on lines 222 and 226 was not immediately clear (see also our answer to comment raised regarding line 102). To avoid any confusion, we have replaced “s” with “ S_M ” to denote the cumulative distribution function in equation (1). This change also aligns better with the common practice of using capital letters for cumulative distribution functions. We have applied this change throughout the manuscript.

Line 276: “We only considered SARs that had at least 50 species”. Arbitrary assumptions can provide interesting “findings”, frequently spurious and arbitrary. The assumptions made in the simulations section, and decisions as this one in line 276, will indeed influence the conclusions, and I don’t think that the title and main claim that EVT explains SAR is valid, the relationship results from the arbitrary assumption that the nearest neighbor of the focal point has normal, Rice or Rayleigh distribution. Observe that for instance Diggle, P.J. 1983. Statistical Analysis of Spatial Point Patterns. New York: Academic Press recommends spatial Poisson randomness, used by Allen and White (reference 51), and many researchers in the field use lognormal (multiplicative pattern, but conforming with power laws signature) instead of normal (additive pattern). All this of course provides a huge variety of regular variation, and establishing connections with EVT seems more a serendipity issue than an essential

finding. Arbitrary assumptions and restrictions have the flavor of the ironical guidance to postgraduate students “get the data that fit your solution”.

“We only considered SARs that had at least 50 species”. Arbitrary assumptions can provide interesting “findings”, frequently spurious and arbitrary

Note that our goal was not to determine the exact shape of a species-area relationship, but rather to illustrate examples of it. Clearly, a SAR with only three species would not provide enough data to visualize the overall pattern. We select a minimum threshold of 50 species to ensure that SARs with a potential triphasic structure had a sufficient number of data points per phase. For instance, it is well known from Spatial Analysis that a common rule of thumb is to have at least 30-50 data points to construct a reliable variogram. This ensures you have enough pairs of points at various distances (lags) to estimate the semivariance for each distance interval. It is not clear why such an arbitrary criterion would necessarily lead to “interesting” findings, given that the general shape of the species-area relationships is well-established.

the relationship results from the arbitrary assumption that the nearest neighbor of the focal point has normal, Rice or Rayleigh distribution.

We would like to clarify that the use of Rayleigh and Rice distributions (for the distances) is not an arbitrary assumption. These distributions naturally result from the underlying assumption of an isotropic bivariate normal distribution. Specifically, the Rayleigh distribution arises when the bivariate normal distribution is centred at the origin, while the Rice distribution arises when it is not. Thus, the only assumption being made here is the use of an isotropic bivariate normal distribution (see below the justification for the use of this distribution in the context of a Neyman-Scott process).

Observe that for instance Diggle, P.J. 1983. Statistical Analysis of Spatial Point Patterns. New York: Academic Press recommends spatial Poisson randomness, used by Allen and White (reference 51), and many researchers in the field use lognormal (multiplicative pattern, but conforming with power laws signature) instead of normal (additive pattern).

In the following, although we do not have access to Diggle (1983), we believe that it is not substantially different from Diggle (2013).

The spatial characteristics of individuals is commonly done using Ripley’s K-function. (It is well established that species often exhibit clustered spatial distributions, meaning individuals tend to be aggregated). Once the nature of these distributions is identified using Ripley’s K-function, simulations of point distributions are often performed using a Neyman-Scott process; e.g., Cressie (1993) or Diggle (2013). This method assumes that clusters are generated from a Poisson process, but, within each cluster, the individuals are distributed according to:

“3. The positions of the offspring relative to their parents are independently and identically distributed according to a d-dimensional density function $f(\cdot)$.” Cressie (1993, page 662)

or

“PCP3 The positions of the offspring relative to their parents are independently and identically distributed according to a bivariate pdf $h(\cdot)$.” Diggle (2013, page 101)

The remaining question is which bivariate distribution should be used. We followed Plotkin et al. (2000) who, using isotropic bivariate normal distributions, successfully simulated realistic spatial distribution of tropical tree species.

It is important to clarify the *distinction between the distribution of the centres of the clusters (modelled as a Poisson process with a given intensity) and the distribution of the individuals within the clusters (modelled with a chosen bivariate distribution)*. In our simulations, we assumed that each species forms a single cluster. This is because we focused on the minimum, and thus only its cluster is relevant to obtaining the SAR. Additionally, using a Poisson process for individual distributions of individuals would result in completely spatial randomness, which does not capture the aggregation patterns commonly observed in real-world data.

used by Allen and White (reference 51), and many researchers in the field use lognormal (multiplicative pattern, but conforming with power laws signature) instead of normal (additive pattern).

Regarding the work by Allen and White, their primary focus is on the transition from Phase II to Phase III and the characteristics of the latter, as they explicitly state: “Here we present a simple mathematical model to account for the increasing slope of the species–area relationship at broad spatial scales”. Not surprisingly, their model is equivalent to our “disk-model”, which applies specifically to Phase III, as we mentioned in line 688:

“Expression S2.1 is equivalent to expression (2) in ⁵¹.” (where reference “51” referred to the work by Allen and White; this reference is now number 52).

Like in our disk-model, their approach is based on two key assumptions:

- “(1) ranges are randomly distributed in space irrespective of size and geographic location and
- (2) ranges are circular in shape.”

It is important to note that Allen and White did not specify the spatial distribution of individuals within the ranges, they simply describe the ranges as “disks”. Thus, the reviewer’s comment about spatial distribution is unclear to us, as the randomness in Allen and White’s work refers specifically to the location of the ranges, not the individuals within them.

As for the statement “many researchers in the field”, the distributions used, and the context where they are used, it would be helpful to have specific references. To our knowledge, one of the few works that explicitly models the spatial distribution of individuals is Plotkin et al. (2000), and they employed a bivariate normal distribution for this purpose.

Cressie, N. (1993). *Statistics for spatial data*. John Wiley & Sons.

Diggle, P. J. (2013). *Statistical analysis of spatial and spatio-temporal point patterns*. CRC press.

Plotkin, J. B., Potts, M. D., Leslie, N., Manokaran, N., LaFrankie, J., & Ashton, P. S. (2000). Species-area curves, spatial aggregation, and habitat specialization in tropical forests. *Journal of theoretical biology*, 207(1), 81-99.

Line 293: “For each taxon and landmass, we obtained 200 SARs” for five landmasses — why there is no indication of the radii of the phases disks?

In our study we chose to report the number of species per area, rather specifying the radii, in line with common practice in ecological studies (see lines 100 to 103 in the original manuscript). Although we did not explicitly provide the radii of the phase disks, we believe the interested reader can easily convert the areas into radii.

Figure 5: I detected daltonism, since I see a green line and the caption speaks of a red line indicating a mean SAR. Why no confidence bands, the diversity of the grey lines showing a widespread of variability?

Thank you for your observation. We corrected the caption. It now reads “the green line indicates”. Regarding the inclusion of confidence bands (or some measure of the dispersion of the data), we tried to balance clarity and complexity in the figures. Adding confidence bands will complicate the visual representation. The SARs presented illustrate considerable variation, which we believe sufficiently conveys the spread in the data.

Lines 327-329: “Here, we showed that extreme value theory offers a comprehensive framework for analysing the SAR, providing a unified explanation for the characteristics of the three phases” As said before, I don’t agree with this bold claim, and more generally I don’t think the main conclusions are properly supported.

We believe we have presented ample evidence to the contrary.

Methods

The Rice and Rayleigh distributions, OK, nothing new but collects information that many readers are unfamiliar with. As said before, some references, such as 34, seem forced. Simulations and Materials The comments on the main body of the paper indicate our reserves on this.

We explained previously why we used reference 34.

Lines 491-492: “1ha, 3ha, and 2ha”. This gives an idea of the length of the radii used, but the absence of that indication is strange. In line 492, it should be 2nd instead of 2rd.

Thank you for pointing out the mistake in line 492.

The exact and asymptotic distributions of the minima Ok, but a minimal indication on the characterization of the domains of attraction of EVT stable laws would provide a better focus on the issues.

We believe this comment is not directly related to lines 491-492, but rather to our discussion of EVT in Supplementary Information A. We thank the reviewer for suggesting an alternative

way to explain the EVT. We added the following paragraphs to the end of Supplementary Information A:

“The previous results can also be stated as follows. For a distribution with cdf $F(r)$, after applying the transformation given by expression S1.2, and assuming the limit exists, the distribution of minima will fall into one of three types: reverse Gumbel, reverse Fréchet or a Weibull. These three distributions constitute the *domain of attraction* for the minima. That is, under the linear transformation, S1.2, the distribution of the minima will converge to one of these three limiting distributions. Moreover, the distributions can be combined into a single distribution call the Generalized Extreme Value (GEV) distribution, which is particularly useful when fitting real data⁸.

The EVT results are somewhat analogous to those of the Central Limit Theorem (CLT) which states that, under certain conditions, the distribution of the sample mean approaches a normal distribution. However, while the CLT deals with a measure of the central tendency of the distribution, EVT focuses on the extreme values of the distribution, of which we are interested here in the minima. Table S1 outlines the distributions used in this work with their respective minimal domains of attraction. For further details on the domains of attraction for maxima and minima of other common distributions, see Castillo et al.⁹, page 207, their table 9.5.

Table S1 | Minimal domains of attraction of the distributions discussed in this manuscript

Parent distribution	Minimal Domain of attraction
Cauchy	Reverse Fréchet
Normal	Reverse Gumbel
Rayleigh	Weibull
Rice	Weibull

“

The Disk Model Aside the comment on radii, I repeat the comment that in nature resources and competition are two main causes for species dispersion.

Indeed, natural resources and competition are two drivers of species dispersion. In our model, however, we are not directly modelling species dispersion, but rather the spatial patterns that emerge from various processes. Our main aim was to illustrate how a simple model can generate the SAR patterns commonly observed.

For instance, along a river I doubt that the disk model is adequate, whereas t can be useful around an isolated lagoon.

SARs are often determined for 2-dimensional regions, e.g., the CTFS 50ha plots (e.g., Condit et al. 1996) or entire countries (Rosenzweig 1995, figure 2.1). Analysing a “SAR” along a river, however, becomes more of a 1-dimensional problem. This is certainly an interesting (and important) question that it is not often addressed, but falls outside the scope of this work. It does, however, deserve to be addressed in future research.

Condit, R., Hubbell, S. P., Lafrankie, J. V., Sukumar, R., Manokaran, N., Foster, R. B., & Ashton, P. S. (1996). Species-area and species-individual relationships for tropical trees: a comparison of three 50-ha plots. *Journal of ecology*, 549-562.

Rosenzweig, M. L. (1995). Species diversity in space and time. Cambridge University Press.

On the other hand, it is known that under or overdispersion indicated by the dispersion index $\text{Var}(X)/E(X)$ is an important parameter, the Poisson distribution with $\text{Var}(XP)/E(XP)=1$ modelling unrestricted randomness (and related to maximum entropy) very much praised, eventually because with only one parameter it is an example of parsimony in fitting (aside from the traditional counting models related to Bernoulli trials, Fisher used a logartmc model, that depending on the parameters may be over or underdispersed, and Engen developed an extended negative binomial model for natural populations).

We believe we have addressed these points in previous responses.

Although this paper deals appropriately with sophisticated mathematical issues, those seem more a way of jumping to conclusions resulting from unproved assumptions than a real explanation of Nature patterns. So, recognizing the mathematical merits but doubting of their appropriateness to provide new insights in the field, and moreover not accepting the main claim that EVT explains SAR, I don't recommend, in the present form, the acceptance of this manuscript (and it is not a question of minor revision, the questions I raise are in the essence of the conception).

We hope to have answered the various issues raised by the reviewer, clarified potential misunderstandings and confusions have been explained, and believe that their main concerns have been addressed.

Report on **Extreme value theory explains the species-area relationship**

by Luís Borda-de-Água, M. Manuela Neves, Luise Quoss, Stephen P. Hubbell, Filipe S. Dias, Henrique M. Pereira

Abstract

line 28: “it increases vary rapidly” it increases very rapidly ?

lines 31-33: “and show that it is the mixture of the distributions of each species individuals’ minimum distances to a starting sampling focal point” “it” refers to what? Surely not to “theory”, or “relationship”, or “EVT”, since none of them can be considered a mixture of distributions.

Line 34: “location of the range”. At this stage what is the location of the range is not clear.

Lines 61-64: “Phase I, the number of species increases rapidly, with a slope approaching 1. Subsequently, for intermediate area sizes, Phase II, the SAR exhibits slower growth, following a power law with an exponent smaller than one. Finally, for large area sizes, Phase III, the SAR experiences rapid growth again, with a slope tending to one¹⁸.” This is obvious, since when a species has been found it is counted and afterwards disregarded. What I would like to understand is the claim about the slopes values, since I think that they must depend on the focal point. As in the last sentence in the abstract the authors say that their results “can be widely applicable to systems that exhibit similar spatial features” I did a rough experiment counting surnames of habitation owners and tenants of habitations in the 32 London boroughs.

[REDACTED]

As expected, results with the focal point in the middle of 34 were different from results with focal point at the frontier of boroughs 6,7,19,20,21, namely because immigration introduced non traditional surnames. In what concerns animal species, something similar must occur with location of resources, as water (which attracts many species, thus providing food to predators, and is the main cause for massive migrations), and moreover there is ample

evidence that the rank-size constraints originate Zipf-Mandelbrot counts (this may be the reason of

Lines 67-68: “In the limiting case where species are represented by only one individual” this is expected in the Zipf theory.

Line 69: “the power law relationship”. Beware of the temptation of power laws, a caveat implied in papers such as Stumpf, M. P. H., and Porter, M. A. (2012). Critical Truths about Power Laws. *Science*, 335(6069), 665–666, in Shalizi, C. R. (2007). So You Think You Have a Power Law? Well Isn't That Special? bactra.org/weblog/491.html or The Econophysics Blog (2006). Tyranny of the Power Law (and Why We Should Become Eclectic). <http://econophysics.blogspot.com/2006/07/tyranny-of-power-law-and-why-we-should.html>

Lines 70-71: “assuming an underlying lognormal species abundance distribution” The use of lognormal models stems out from the fact that it mimics the linear signature of power laws in log-log plots, a feature explained in the pioneer paper Belevitch, V. (1959). On the Statistical Laws of Linguistic Distributions. *Annales de la Société Scientifique de Bruxelles*, Ser. I, 73, 310–326. <http://www.csl.sri.com/users/neumann/belevitch.pdf>. Once again, confounding simple serendipity with the essence of the question.

Line 102: “can be defined as an equally weighted sum”. I don't see a justification to consider “equally”.

Line 111-112: “... Gumbel distributions⁷⁻⁹ A comment on the many references (and I didn't check most of them): what is the reason for so many references when a single one would be enough? For instance in what regards EVT, aside from the books 7, 8, 9, 26, at the end of the supplementary materials the initial papers by Fréchet (48) and Fisher and Tippet (49), and also a paper by Gumbel (50) are listed. Speaking of references, for instance in references 49 and 50 is used a “in” whereas the publication is a journal, not an edited book. Another thing on the references list, 17, 29 and 38 use as authors [first author] et al., which I consider inappropriate. I also consider that references seem exaggerated, what is for instance the relevance of a package (34) for transforming characteristic functions (that are relevant for sums but not for extremes) in CDF's or PDF's?

Lines 124-125: “distributed according to bivariate normal distributions with scale parameter $p=1$ and randomly uniform centres.” Those arbitrary assumptions will unduly narrow down the EVT models to Weibull-0.5 and to Gumbel (although the slow and regularly varying left tail variation characterization of domains of attraction of stable EVT laws are never referred). Observe that in Lines 927- 931 the goodness of fit of the normal model is rejected.

Line 141: “cases an discuss” cases and discuss

Lines 154-155: “assuming the individuals' locations to be isotropic bivariate Cauchy distributions”. The Cauchy is additive stable with index 1, so due to the similarity of characterizations of the domains of attraction of stable laws in the additive and in the EVT contexts it is in the EVT domain of attraction of the Fréchet 1, so the claim that the “results were qualitatively similar” is vague.

Line 199: what is the meaning of $\xi \lesssim 0$?

Lines 222 and 226: $s_{P(I-II)} = s_{P(I-II)}/S_T$, $s_{P(II-III)} = s_{P(II-III)}/S_T$ S_T , the minima of all species (line 103) is 1?

Line 276: “We only considered SARs that had at least 50 species”. Arbitrary assumptions can provide interesting “findings”, frequently spurious and arbitrary. The assumptions made in the simulations section, and decisions as this one in line 276, will indeed influence the conclusions, and I don’t think that the title and main claim that EVT explains SAR is valid, the relationship results from the arbitrary assumption that the nearest neighbor of the focal point has normal, Rice or Rayleigh distribution. Observe that for instance Diggle, P.J. 1983. *Statistical Analysis of Spatial Point Patterns*. New York: Academic Press recommends spatial Poisson randomness, used by Allen and White (reference 51), and many researchers in the field use lognormal (multiplicative pattern, but conforming with power laws signature) instead of normal (additive pattern). All this of course provides a huge variety of regular variation, and establishing connections with EVT seems more a serendipity issue than an essential finding. Arbitrary assumptions and restrictions have the flavor of the ironical guidance to postgraduate students “get the data that fit your solution”.

Line 293: “For each taxon and landmass, we obtained 200 SARs” for five landmasses — why there is no indication of the radii of the phases disks?

Figure 5: I detected daltonism, since I see a green line and the caption speaks of a red line indicating a mean SAR. Why no confidence bands, the diversity of the grey lines showing a widespread of variability?

Lines 327-329: “Here, we showed that extreme value theory offers a comprehensive framework for analysing the SAR, providing a unified explanation for the characteristics of the three phases” As said before, I don’t agree with this bold claim, and more generally I don’t think the main conclusions are properly supported.

Methods

The Rice and Rayleigh distributions, OK, nothing new but collects information that many readers are unfamiliar with. As said before, some references, such as 34, seem forced.

Simulations and Materials The comments on the main body of the paper indicate our reserves on this.

Lines 491-492: “1ha, 3ha, and 2ha”. This gives an idea of the length of the radii used, but the absence of that indication is strange. In line 492, it should be 2nd instead of 2^d.

The exact and asymptotic distributions of the minima Ok, but a minimal indication on the characterization of the domains of attraction of EVT stable laws would provide a better focus on the issues.

The Disk Model Aside the comment on radii, I repeat the comment that in nature resources and competition are two main causes for species dispersion. For instance, along a

river I doubt that the disk model is adequate, whereas it can be useful around an isolated lagoon. On the other hand, it is known that under or overdispersion indicated by the dispersion index $\text{Var}(X)/E(X)$ is an important parameter, the Poisson distribution with $\text{Var}(X_P)/E(X_P)=1$ modelling unrestricted randomness (and related to maximum entropy) very much praised, eventually because with only one parameter it is an example of parsimony in fitting (aside from the traditional counting models related to Bernoulli trials, Fisher used a logartmc model, that depending on the parameters may be over or underdispersed, and Engen developed an extended negative binomial model for natural populations).

Although this paper deals appropriately with sophisticated mathematical issues, those seem more a way of jumping to conclusions resulting from unproved assumptions than a real explanation of Nature patterns. So, recognizing the mathematical merits but doubting of their appropriateness to provide new insights in the field, and moreover not accepting the main claim that EVT explains SAR, I don't recommend, in the present form, the acceptance of this manuscript (and it is not a question of minor revision, the questions I raise are in the essence of the conception).